# Single-cell normalization and association testing unifying CRISPR screen and gene co-expression analyses with Normalisr

Lingfei Wang [1,2,3 ✉]

Single-cell RNA sequencing (scRNA-seq) provides unprecedented technical and statistical potential to study gene regulation but is subject to technical variations and sparsity. Furthermore, statistical association testing remains difficult for scRNA-seq. Here we present Normalisr, a normalization and statistical association testing framework that unifies single-cell differential expression, co-expression, and CRISPR screen analyses with linear models. By systematically detecting and removing nonlinear confounders arising from library size at mean and variance levels, Normalisr achieves high sensitivity, specificity, speed, and generalizability across multiple scRNA-seq protocols and experimental conditions with unbiased p-value estimation. The superior scalability allows us to reconstruct robust gene regulatory networks from trans-effects of guide RNAs in large-scale single cell CRISPRi screens. On conventional scRNA-seq, Normalisr recovers gene-level co-expression networks that recapitulated known gene functions.

[1] Broad Institute of MIT and Harvard, Cambridge, MA, USA. [2] Center for Computational and Integrative Biology, Massachusetts General Hospital, Boston, MA, USA. [3] Molecular Pathology Unit and Center for Cancer Research, Massachusetts General Hospital Research Institute, Charlestown, MA, USA. ✉email: lwang55@mgh.harvard.edu

Understanding gene regulatory networks and their phenotypic outcomes forms a major part of biological studies. RNA sequencing (RNA-seq) has received particular popularity for systematically screening transcriptional gene regulation and co-regulation. Single-cell RNA-seq (scRNA-seq) provides a unique glance into cellular transcriptomic variations beyond the capabilities of bulk technologies, enabling analyses such as single-cell differential expression (DE)[1], co-expression[2,3], and causal network inference[4,5] on cell subsets at will. Regarding and manipulating each cell independently, especially in combination with CRISPR (clustered regularly interspaced short palindromic repeats) technology[6,7], scRNA-seq can overcome the major limitations in sample and perturbation richness of bulk studies, at a fraction of the cost.

However, cell-to-cell technical variations and low read counts in scRNA-seq remain challenging in computational and statistical perspectives. A comprehensive benchmarking found that single-cell-specific attempts in DE could not outperform existing bulk methods such as edgeR[8]. Generalized linear models (e.g., refs. [1,9]) are difficult to generalize to single-cell gene co-expression, leaving it susceptible to expression-dependent normalization biases[10,11]. Single-cell CRISPR screen analysis can be regarded as multivariate DE, but existing methods (e.g., scMageck[12] and SCEPTRE[13]) take weeks or more on a modern dataset and do not account for off-target effects. These challenges limit the accuracy, efficiency, scalability, and flexibility of single-cell data analysis and the downstream biological discovery.

A potential unified framework for single-cell DE, co-expression and CRISPR analysis is a two-step normalization–association inferential process (Fig. 1a). The normalization step (e.g., sctransform[14], bayNorm[15], and Sanity[16]) removes confounding technical noises from raw read counts to recover the biological variations. The subsequent linear association step has been widely applied in the statistical analyses of bulk RNA-seq and microarray co-expression[17], genome-wide association studies[18,19], expression quantitative trait loci[20], and causal network inference[21,22]. Linear association testing presents several advantages: (i) exact $P$ value estimation without permutation, (ii) native removal of covariates (e.g., batches, house-keeping programs, and untested guide RNAs (gRNAs)) as fixed effects, (iii) robustness against read count distribution distortions with enough (>100) cells[23], and (iv) computational efficiency. However, existing normalization methods were designed primarily for clustering. Sensitivity, specificity, and effect size estimation in association testing are left mostly uncharted.

Here we present Normalisr, a normalization–association framework for statistical inference of gene regulation and co-expression in scRNA-seq (Fig. 1a). The normalization step

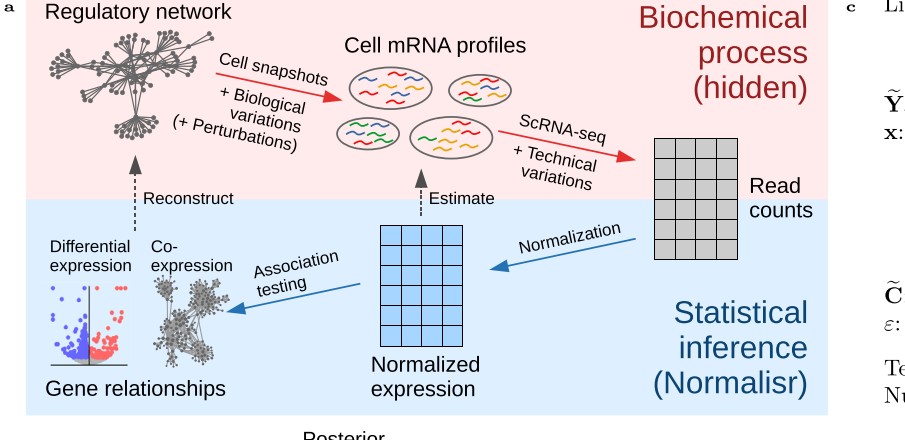

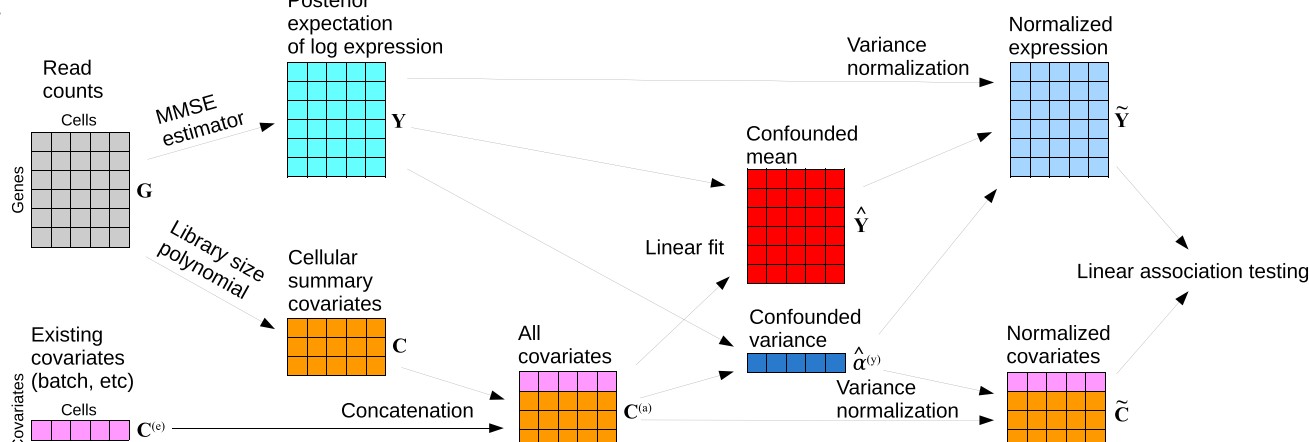

**Figure 1 Normalisr overview. a** Schematics of biochemical and inferential processes (top to bottom). Normalisr aims to remove technical variations and normalize raw read counts to estimate the pre-measurement mRNA abundances. Then they can be directly handled by conventional statistical methods, such as linear models, to infer gene regulation and to unify different analyses. **b** Normalization step (in **a**) of Normalisr (left to right). Normalisr starts by computing the expectation of posterior distribution of mRNA log proportion in each cell with MMSE. Meanwhile, Normalisr appends existing covariates with nonlinear cellular summary covariates. Normalisr then normalizes expression variance by linearly removing the confounding effects of covariates on log variance. The normalized expression and covariates are ready for downstream association testing with linear models, such as differential and co-expression (in **c**). **c** Association testing step (in **a**) of Normalisr, which uses linear models to test gene differential and co-expression.

estimates the pre-measurement mRNA frequencies from the scRNA-seq read counts (e.g., unique molecular identifiers (UMIs)) and regresses out the nonlinear effect of library size on expression variance (Fig. 1b). The association step utilizes linear models to unify case–control DE, single-cell CRISPR screen analysis as multivariate DE, and gene co-expression network inference (Fig. 1c). We demonstrate Normalisr's applications in two scenarios: gene regulation screening from pooled CRISPR interference (CRISPRi) CRISPR droplet sequencing (CROP-seq) screens and the reconstruction of transcriptome-wide co-expression networks from conventional scRNA-seq.

## Results

**Normalisr overview**. For UMI-based assays[10,24], we regarded scRNA-seq as a binomial (approximate of multinomial) mRNA sampling process $Binom(n, p)$ without zero inflation. Given the sequencing read counts of all genes as $n$ and of one particular gene as $k$ in each cell, the key question is to construct an estimator for log relative expression of this gene as $\ln p$. Since the uniformly minimum variance unbiased estimator for $\ln p$ does not exist (Supplementary Information), we turned to its Bayesian analog—the minimum mean square error (MMSE) estimator. Briefly, this Bayes estimator computes the posterior expectation of $\ln p$ for each gene in each cell (namely, Bayesian log expression) based on a non-informative (uniform) prior for the binomial mRNA sampling process ("Methods" and Supplementary Information). This estimator naturally addresses two drawbacks of the conventional normalization with $\log(CPM + constant)$. First, it avoids the artificial introduction and choice of constant. Second, zero-read genes are indifferent after logCPM normalization regardless of the total read count in each cell but are properly assigned with higher relative expression in lowly sequenced cells by MMSE to account for pool size differences. We also intentionally avoided imputations that rely on information from other genes that may introduce spurious gene inter-dependencies or from other cells that assume whole-population homogeneity and consequently risk reducing sensitivity (see below).

To account for potential technical confounders, we introduced two previously characterized cellular covariates: the number of zero-read genes[1] and log total read count[14]. These two covariates comprise the complete set of unbiased cellular summary statistics up to first order (as $L^0$ and $L^1$ norms of cell read count, respectively, Supplementary Information) without artificial selection of gene subsets. Restricting to unbiased covariates minimizes potential interference with true co-expression networks. The number of zero-read genes also provides extra information in the read count distribution for each cell and allows for a regression-based automatic correction for distribution distortions. This approach aims for the same goal as quantile normalization but is not restricted by the low read counts in scRNA-seq.

We modeled nonlinear confounding effects on gene expression with Taylor polynomial expansion. We iteratively introduced higher-order covariates to minimize false co-expression on a synthetic co-expression-free dataset, using forward stepwise feature selection that is restricted to series expansion terms whose all lower-order terms had already been included. The set of nonlinear covariates were pre-determined as such and then universally introduced for all datasets and statistical tests. We first linearly regressed out these covariates' confounding effects at the log variance level[25]. Their mean confounding was accounted for in the final hypothesis testing of linear association (Fig. 1b, c). Finally, $P$ values were computed from Beta distribution of $R^2$ in exact conditional Pearson correlation test (equivalent of likelihood ratio test) without permutation. Because the conditional Pearson correlation test was performed between the target

variables of interest, the outcome would not be affected by potential correlations within covariates themselves.

**Normalisr detected nonlinear technical confounders with restricted forward stepwise selection**. We downloaded the "UPR Perturb-seq experiment" dataset of UMI read count and gRNA assignment matrices of K562 cells[6] based on 10× Genomics to determine the nonlinear confounding of cellular summary covariates. Perturb-seq was proposed to efficiently quantify the effects of multiple CRISPR perturbations on the transcriptome by detecting gene expression and CRISPR gRNA occurrence in the same single cells. Therefore, this dataset can provide positive controls in DE for our sensitivity evaluation of different methods afterwards.

For nonlinear confounding detection, we generated a synthetic null scRNA-seq dataset to mimic the gRNA-free K562 cells but without any co-expression, with log-normal and multinomial distributions, respectively, for biological and technical variations (Fig. S1 and "Methods"). (For exact gene, cell, and gRNA counts here and onwards, see Supplementary Data 1.) This simplified scenario simulates the technical variations from heterogeneous mRNA sampling rates in multinomial sequencing process on homogeneous cells. It recapitulated the technical confounding from read count sparsity on conventional logCPM normalization, as demonstrated by the 0-biased $P$ values of Pearson correlation between genes (Fig. S2).

Using the cellular summary covariates as the basis features and optimizing toward a uniform distribution of (conditional) Pearson co-expression $P$ values, we confirmed that both the log total read count and the number of zero-read genes in a cell strongly confound single-cell mRNA read count (Fig. S3). Moreover, we identified strong nonlinear confounding from the square of log total read count. Together, these three extra covariates were sufficient to recover the uniform null distribution of co-expression $P$ values. These predetermined nonlinear cellular covariates allowed Normalisr to perform hypothesis testing without any parameter or permutation in subsequent evaluations and applications.

**Evaluations in single-cell DE and co-expression**. We performed a wide range of evaluations with the Perturb-seq dataset on Normalisr, other normalization methods including sctransform[14], bayNorm[15], and Sanity[16], imputation methods including MAGIC[26], DCA[27], DeepImpute[28], scDoc[29], DrImpute[30], and EnImpute[31], and DE methods including Seurat (with default Wilcoxon test)[32], edgeR, and MAST[8] ("Methods"). We partitioned the 4622 cells that did not detect any gRNA into two random groups 100 times to evaluate the null $P$ value distribution of single-cell DE. This provides better homogeneity than using cells with non-targeting control (NTC) gRNAs because they may have unknown and unintended targets (see below). To detect expression-dependent null $P$ value biases, we grouped genes into ten equally sized subsets from low to high expression (proportion of expressed cells) and compared their null $P$ values against the uniform distribution with Kolmogorov–Smirnov (KS) test ("Methods"). Normalisr recovered uniform distributions of null DE $P$ values at all expression levels, as did other normalization methods and some imputation methods that were followed by the same linear model (Fig. 1c) and Seurat (Figs. 2a, b and S4). In contrast, edgeR and MAST, the best single-cell DE methods benchmarked in ref. [8], had expression-dependent, 0- or 1-biased null $P$ values. Normalisr was additionally much faster than most other methods tested, and over 2000 times than Seurat, edgeR, and MAST (Fig. 2c).

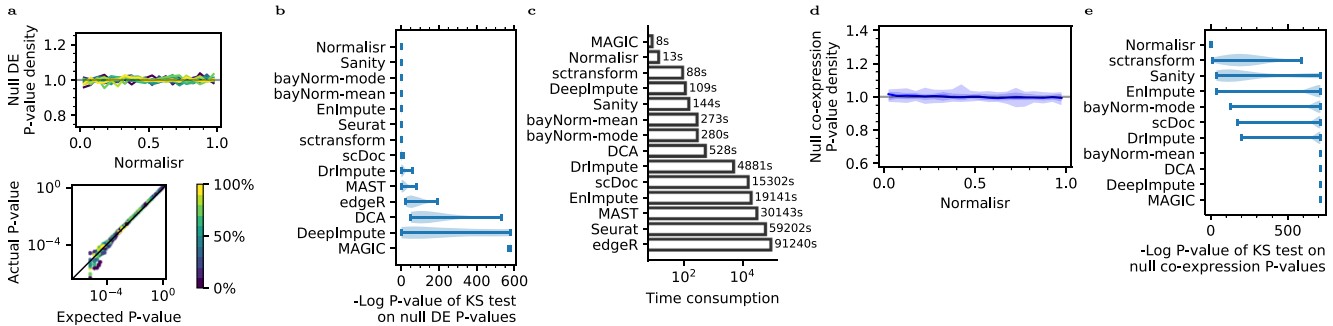

**Figure 2 Normalisr achieved high specificity and speed in single-cell DE and co-expression. a** Normalisr had uniformly distributed null *P* values in single-cell DE (top *X* and bottom *Y*) as shown by histogram density (top *Y*) and quantile–quantile (Q-Q) plot (bottom). Q-Q plot shows the false positive rate (*Y*) at different cutoffs (*X*). Genes were evaluated in 10 equally sized (≈976) and separately colored bins stratified by expression (proportion of expressed cells). Gray line indicates the expected uniform distribution for null *P* value. **b** Normalisr recovered uniformly distributed null *P* values for DE, as measured by the violin plot of *P* values of KS test (*X*) on null *P* values separately for each gene bin (in **a**). Bars show extrema. **c** Normalisr was much faster than most other methods. **d** Normalisr had uniformly distributed null *P* value in single-cell co-expression (*X*) from synthetic data, as shown by histogram density (*Y*). Genes were split into ten equal bins from low to high expression. The null *P* value distribution of co-expression between each bin pair formed a separate histogram curve. Central curve shows the median of all histogram curves. Shades show 50, 80, and 100% of all histograms. Gray line indicates uniform distribution. **e** No other method tested could recover uniformly distributed null *P* values for co-expression, as measured by the violin plot of *P* values of KS test (*X*) on null *P* values separately for each gene bin pair (in **d**). Bars show extrema. This figure uses two-sided raw *P* values.

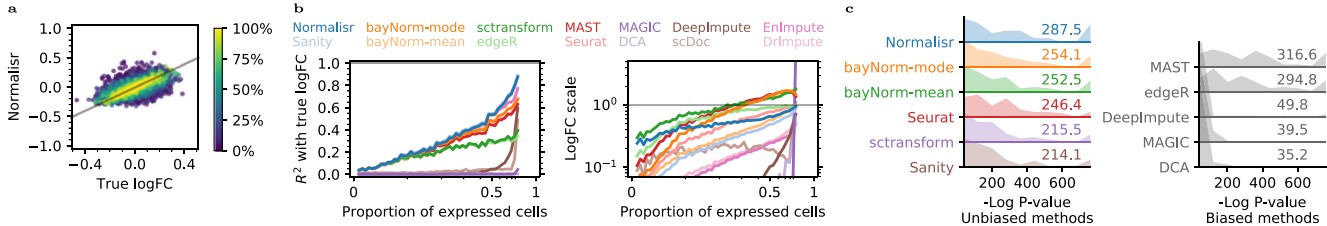

**Figure 3 Normalisr achieved high sensitivity and high logFC estimation accuracy in single-cell DE. a** Normalisr accurately recovered logFCs (*Y*) when compared against the synthetic ground-truth (*X*) for genes from low to high expression (color). Gray line indicates *X* = *Y*. **b** Normalisr accurately recovered logFCs with low variance (left, *Y* as *R²*) and low bias (right, *Y* as regression coefficient) when evaluated against synthetic ground-truth with a linear regression model separately for genes grouped from low to high expression (*X*) on logFC scatter plots (**a** and Fig. S5). Horizontal gray line indicates optimal performance. The nonlinear scale in *X* distributes genes uniformly in ascending order so an average *Y* reflects average performance over genes. **c** Normalisr was the most sensitive among unbiased DE methods (left, see Fig. 2b) in the CRISPRi Perturb-seq experiment. Histogram shades show distributions (*Y*) of negative log *P* values (*X*) of the targeted gene between KD and unperturbed cells (positive controls with varying strengths). Numbers indicate mean values. Stronger *P* values from unbiased methods indicate higher sensitivity. Biased methods (right) are listed in gray.

To detect potential incomplete removal of technical confounding, we compared the distribution of co-expression *P* values from the synthetic null dataset for normalization and imputation methods followed by the same linear model for hypothesis testing (Fig. 1c and "Methods"). Normalisr recovered uniformly distributed *P* values irrespective of expression and better than log(CPM+1) with or without covariates (Figs. 2d, e and S2), suggesting the necessities of both the Bayesian log expression and the nonlinear cellular summary covariates. No other method could fully account for technical confounding and recover uniform null *P* value distributions. Imputation methods may inflate gene–gene associations because of their inherent reliance on gene relationships. Normalisr accounted for technical confounding and correctly recovered the absence of co-expression.

To evaluate the effect size estimation in terms of log Fold-Change (logFC), we used synthetic datasets from null co-expression and from simulations by SymSim[33] and Splatter[34]. These simulations mimicked the real dataset and also recorded the biological ground-truths of cell mRNA profiles or logFCs (Fig. 1a and "Methods"). We decomposed logFC estimation errors of each method into bias and variance with linear regression against the ground-truth logFC. Bias indicates the overall under- or over-estimation errors in logFC scale as the

deviation of regression coefficient from unity. Variance represents uncertainties in the estimation and was quantified with *R²* of the regression. Although these simulation methods vary by model and by similarity to the real dataset (Fig. S1), Normalisr consistently obtained one of the lowest biases and variances across all expression levels (Figs. 3a, b and S5). BayNorm underestimated the logFCs of lowly expressed genes, and Sanity underestimated the logFCs regardless of expression. Their prior distributions aggregate information across cells under the implicit assumption of population homogeneity, which may over-homogenize gene expression and consequently underestimate logFCs, especially for lowly expressed genes whose fewer reads possessed less information to overcome the prior. On the other hand, edgeR recovered the most accurate logFCs for moderately and highly expressed genes but was susceptible to large bias for lowly expressed genes and large overall variance. In summary, Normalisr accurately estimated logFCs with low bias and variance for genes across all expression levels.

We then assessed the sensitivity or statistical power of different methods in detecting differentially expressed genes in scRNA-seq. In the absence of a high-quality gold standard dataset, we focused on the intended CRISPRi effects on the expression of targeted gene between cells infected by the corresponding single gRNA against uninfected cells (excluding cells infected with any other

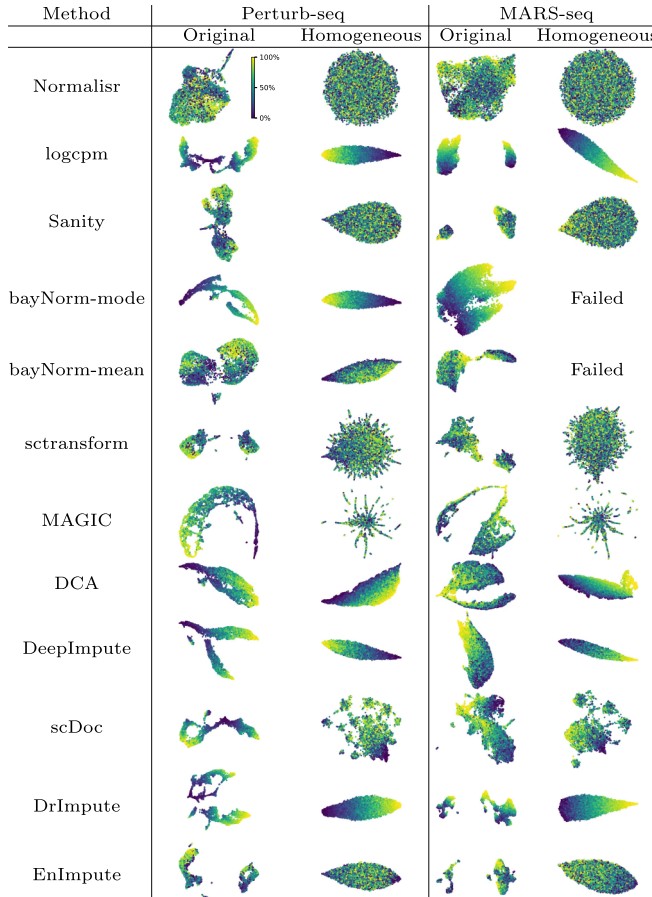

**Figure 4 Normalisr removed library size confounding bias in UMAP embedding on Perturb-seq and MARS-seq datasets as well as their synthetic datasets.** On the original datasets, cell coordinates from Normalisr's normalized expression did not have any directional dependency on the proportion of expressed genes. Synthetic homogeneous datasets from null co-expression contained homogeneous cells whose uniform relationships could only be captured by Normalisr with a disc in UMAP embedding. Color indicates cell ranking in the proportion of expressed genes.

gRNA) as positive controls. Although CRISPRi efficiency varies, a more sensitive method should find a stronger DE *P* value on average with the same data. Among the successfully finished methods with unbiased *P* values (Fig. 2b), which are necessary to reflect sensitivity, Normalisr obtained the most significant *P* values (Fig. 3c). Sanity, sctransform, and Seurat suffered sensitivity losses. Imputation methods lost the major effects of CRISPRi.

The above evaluations were based on and combined from a wide range of effective cell/sample counts, which were constrained and modulated by the size of the minor group among the two compared in DE. The number of cells in the minor group varied greatly across 100 random groupings in null DE evaluation (from 242 to 2305 for Figs. 2a–c and S4), across different gRNAs in sensitivity evaluation (from 185 to 1480 for Fig. 3c), and across 100 random groupings in logFC estimation evaluation (from 153 to 3911 for Fig. 3a, b) and were chosen at random four times in SymSim and Splatter for logFC estimation evaluation (from 519 to 3927 for Fig. S5b, c). The comprehensive design covered all major evaluation metrics in DE and emulated potential applications such as marker gene detection and large-scale screens with different cell count imbalances. Normalisr's consistent

performance over various effective cell counts suggested reliable performance in diverse settings.

Our evaluation results were also reproducible on a MARS-seq dataset[35] of dysfunctional T cells from frozen human melanoma tissue samples (Figs. S1, S3, and S6–S8) that is further explored for co-expression later. This dataset utilized antibody-based fluorescence-activated cell sorting prior to well-based scRNA-seq, with a much smaller library size than the Perturb-seq dataset (containing 2409/5,132 expressed genes and 9784/32,625 UMIs per cell among the top 100 cells with the most UMIs). Normalisr's superior performance was consistent across multiple scRNA-seq platforms, library sizes, cell counts, and sample conditions.

To visually compare different normalization and imputation methods on their technical bias removal effects on cell population structure, we performed UMAP to embed cells onto low dimensions from normalized or imputed transcriptome matrix ("Methods"). Normalisr successfully removed library size confounding bias, showing no cell coordinate dependency on the proportion of expressed genes (Fig. 4). Moreover, only Normalisr could fully remove technical confounding and recover the absence of cell population structure on the synthetic null datasets of homogeneous cells. Normalisr successfully removed technical bias on low dimensions.

In total, Normalisr provides a normalization framework with improved sensitivity, specificity, and efficiency than existing bulk and single-cell methods to allow linear association testing for single-cell differential and co-expression that removes technical confounding bias and is consistent across multiple experimental conditions.

**Normalisr reduced the false positives from gRNA cross-associations in high-multiplicity of infection (MOI) single-cell CRISPR screens.** High-MOI CRISPRi systems are highly efficient screens for gene regulation, gene function[6,36], and regulatory elements[7]. However, gRNA–mRNA associations may have high false positive rates (FPR) when confounded by gRNA cross-associations, such as from the competition between gRNAs for dCas9 or for limited read counts, or from selective advantage due to gRNA presence. Regardless of origin, gRNA cross-associations can incur false positive upregulation and downregulation of genes regulated by other gRNAs or their target genes (Fig. 5a).

To understand the frequency of gRNA cross-associations, we utilized one of the largest enhancer-screening CROP-seq datasets to date[7] of 207,324 K562 cells post quality control (QC), containing 13,186 gRNAs and 10,877 genes. Over 98.7% cells detected more than one gRNA averaging at 30 gRNAs per cell. Over 20% cells contained at least 1 of the 101 NTCs. Cells already containing gRNAs were indeed less likely to include another gRNA (Fig. 5b, Methods), significantly deviating from a random gRNA assignment. This was reproducible in a small-scale pilot screen of the same study with 3,117 gRNAs and 47,650 cells averaging at 18 gRNAs per cell, including the same NTCs and transcription start site (TSS)-targeting gRNAs but fewer enhancer-candidate-targeting gRNAs.

We then used NTCs as negative controls to estimate the elevated FPR from gRNA–gene associations. A competition-naive DE analysis using Normalisr that disregarded other, untested gRNAs lead to a 4.8% overall FPR among NTCs (Figs. 5c and S9, Storey's method[37]). Notably, the FPR was significantly higher among genes whose TSSs were targeted by another gRNA (Fig. 5c, 10.3% than 4.6% for other genes). This supported our hypothesis that false positives were mediated by TSS-targeting gRNAs, because indirect effects through the targeted genes are weaker and harder to detect (Fig. 5a).

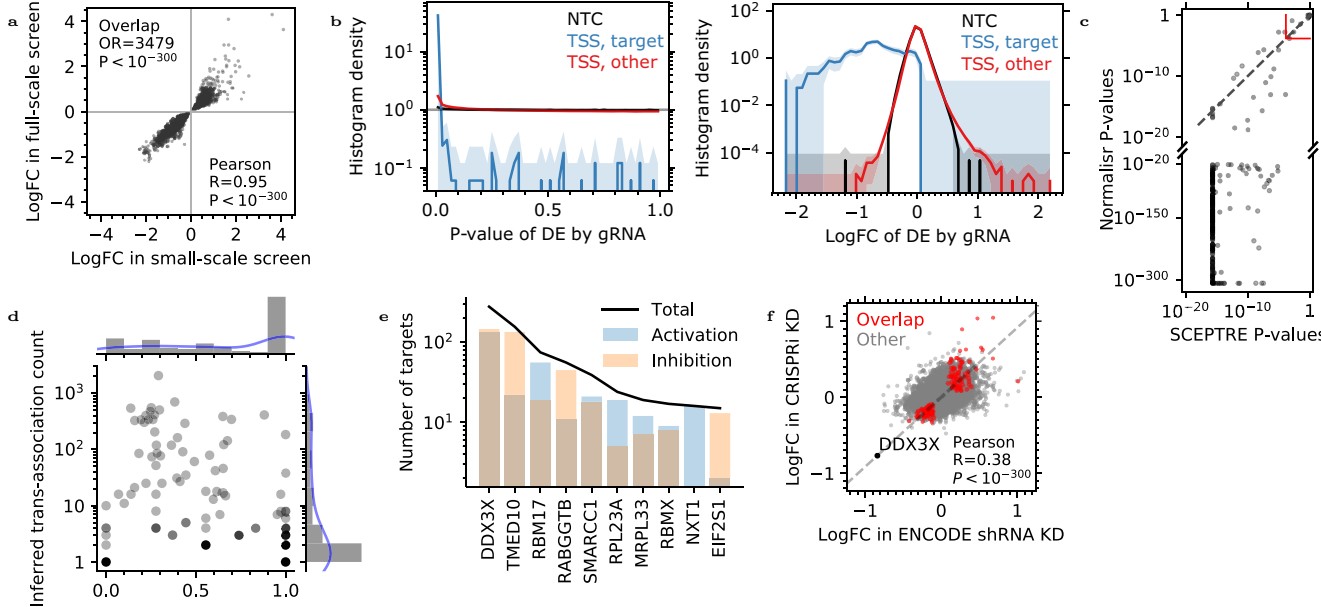

**Figure 5 Normalisr reduced the false positives from gRNA cross-associations in high-MOI single-cell CRISPR screens. a** Example scenario of false positives of gRNA–gene associations (dashed) arising from negative gRNA cross-associations and true regulation (solid) for genes targeted directly by the effective gRNA or indirectly through the targeted genes (other gene). **b** Detection of different gRNAs were anti-correlated across cells in terms of one-sided hypergeometric $P$ values (left, $X$) and odds ratios (right, $X$) between gRNA pairs. Empirical PDFs and CDFs are shown in blue (left $Y$) and red (right $Y$), respectively. **c** Ignoring untested gRNAs increased FPR, as shown by density histogram ($Y$) of CRISPRi DE two-sided $P$ values from NTC gRNAs ($X$). $P$ value histograms and FPRs were computed separately with competition-naive (ignoring untested gRNAs) and competition-aware (regarding untested gRNAs as covariates) methods and separately for genes targeted by positive control gRNAs at the TSS and other genes. Gray line indicates the expected uniform distribution for null $P$ value. Shades indicate absolute errors estimated as $2\sqrt{N+1}$, where $N$ is the count in each bin. **d** Competition-naive DE inflated the number of significant gRNA–gene associations relative to the competition-aware method ($Y$) at different nominal Q-values ($X$).

**Figure 6 Normalisr detected robust and specific gene regulation from high-MOI CRISPRi systems. a** Guide RNA–gene associations were highly reproducible among 1857 significant regulations (dot) between the logFCs in small-scale ($X$) and full-scale ($Y$) CRISPRi screens. **b** Normalisr uncovered gene regulations as secondary effects of TSS-targeting gRNAs, which was significantly stronger than NTCs in terms of histograms ($Y$) of DE $P$ value (left, $X$) and logFC (right, $X$). Gray lines indicate the expected uniform distribution for null $P$ value. Shades indicate absolute errors estimated as $2\sqrt{N+1}$, where $N$ is the count in each bin. **c** Normalisr is more sensitive than SCEPTRE on positive control CRISPRi repressions (dots) in terms of $P$ values. Dashed gray line indicates equal sensitivity. Solid red line is the significance cutoff by Normalisr or SCEPTRE (Bonferroni $P < 0.05$). **d** Over half of significant *trans*-associations were potential off-target effects at $Q < 0.05$. The putative off-target rate ($X$) among the number of inferred *trans*-associations ($Y$) is shown for each gene (dot) that is directly targeted by two gRNAs at the TSS. **e** Numbers of inferred targets ($Y$) of top regulators ($X$) showed different activation and inhibition preferences. **f** Gene DE logFC by *DDX3X* knock-down in CRISPRi screen ($Y$) were confirmed with that from ENCODE shRNA knock-down ($X$). Overlapped targets are highlighted in red (OR = 1.16, hypergeometric $P = 0.01$). Dashed line passes through *DDX3X* and the origin. This figure uses two-sided raw $P$ values.

Linear models can account for other, untested gRNAs as additional covariates in a competition-aware model. Inspired by ref. [38], we also reduced the covariates, when >10,000, to their top 500 principal components (PCs) as an efficient heuristic solution, which was evaluated to be robust on the small-scale screen (Fig. S10 and "Methods"). This allowed Normalisr to statistically test all *cis*- and *trans*-regulations between every gRNA–gene pair in the full-scale screen within a day on a 64-core computer. In comparison, scMAGeCK failed to finish within 2 weeks and SCEPTRE, Seurat, edgeR, and MAST are projected to need over a

year. Normalisr was uniquely efficient for modern-scale single-cell CRISPR screens.

Normalisr successfully recovered NTCs' improved, near-uniform $P$ value distribution independent of whether the gene was targeted at the TSS (Figs. 5 and S9, overall FPR = 1.5%). On the contrary, the competition-naive method under-estimated the false discovery rate (FDR, controlled by Q-values in this paper), reporting over 15 times the gRNA–gene associations for NTC gRNAs and over 4 times for candidate enhancer-targeting gRNAs at nominal $Q \le 0.2$ than the competition-aware method (Fig. 5d).

Normalisr could account for gRNA cross-associations and reduce the consequent false positives in gRNA effects.

**Normalisr reconstructed causal gene regulatory networks from high-MOI CROP-seq screens.** In order to reconstruct causal gene regulatory networks, we first verified the quality of the dataset and the analyses in multiple aspects. First, to validate the reproducibility of highly significant gRNA–gene associations (Bonferroni $P \leq 0.05$ in each screen), we performed the same inference on the small-scale screen. We found major overlap between the two screens, with highly correlated logFCs and all effect directions matched (Fig. 6a). We also detected unexpected associations of NTCs with gene expression that were consistent between screens ($P < 3 \times 10^{-4}$ in each screen and total Bonferroni $P < 0.1$, Supplementary Data 2). The affected genes were uniformly repressed suggesting potential off-targeting, with the exception of *ZFC3H1* that plays a major role in RNA degradation[39]. Normalisr's regulation inference was reproducible across the screens.

gRNAs targeting TSS were regarded as positive controls in ref. [7], and indeed we found 95% (703/738 at $Q \leq 0.05$) to significantly inhibit the expression of targeted genes with varying efficiencies (Fig. 6b). In comparison with SCEPTRE that was regarded most sensitive[13], Normalisr was even more sensitive at gene level (Fisher's method) and could identify most (348/357) of these positive controls at a stronger $P$ value (Fig. 6c). This lead remained apparent even after considering SCEPTRE's $P$ value lower bound from resampling.

With two gRNAs targeting the same TSS, we quantified the proportion of off-target effects as the proportion of significant associations ($Q \leq 0.05$) between the weaker gRNA (in association with the targeted gene) and its *trans*-genes (over 1 Mbp away from the targeted site or on different chromosomes) that were highly insignificant for the stronger gRNA ("Methods"). We found over half of the inferred *trans*-associations were likely off-target effects on average (Fig. 6d). The mean off-target rate was significantly reduced at a more stringent threshold ($Q \leq 10^{-5}$, Fig. S11), suggesting that the putative off-target effects were on average weaker than genuine *trans*-associations.

Despite the dataset's original design as an enhancer screen where each enhancer-candidate-targeting gRNA tests one regulatory element, we repurposed it as a gene regulation screen where each TSS-targeting gRNA tests all the active transcriptomic regulations by the targeted gene. The search for gene regulations is a long standing question on its own in systems biology, which also offers molecular mechanistic insights[40]. Here, based on causal inference[21,22,41], we reconstructed a gene regulatory network by searching for TSS-targeting gRNA → targeted gene → *trans*-gene relationships using gRNA presence as an instrumental variable. This requires simultaneous satisfaction of causal dependencies: (i) TSS-targeting gRNA → targeted gene, (ii) TSS-targeting gRNA → *trans*-gene, and (iii) TSS-targeting gRNA → *trans*-gene only through the targeted gene. Since we already statistically tested (i) and (ii) for all gRNAs and genes above, we then tested and rejected two major violations of (iii)—mediation through nearby genes and gRNA off-target effects—by excluding gRNAs that inhibited another gene within 1 Mbp from the TSS and gene regulations irreproducible across gRNAs targeting the same TSS or across screens. In total, we recovered 833 high-confidence putative gene regulations (all four $Q \leq 0.2$, "Methods") that formed a gene regulatory network (Supplementary Data 3).

Some of the top identified regulators exhibited strong preferences in upregulation or downregulation of other mRNAs (Fig. 6e and Supplementary Data 4). *TMED10*, a.k.a. p24δ1, is responsible for selective protein trafficking at the endoplasmic reticulum (ER)–Golgi interface[42], and indeed its inhibition upregulated ER- and Golgi-localized genes. Inhibition of *RBM17*, part of the spliceosome and essential for K562 cells

(ranked at 3.8% in CRISPR knock-out screen[43]), modulated cell respiration gene expression. *RABGGTB* upregulated vesicle-localized genes, in agreement with its role as a Rab geranylgeranyl transferase subunit[44].

We cross-validated Normalisr's DE against bulk RNA-seq of K562 cells with *DDX3X* knock-down and control short hairpin RNA vectors from the ENCODE consortium[45,46] using edgeR ($Q \leq 0.05$). Despite its limited sample size (two each) and differently engineered cell lines, we observed an agreement in logFC and significant overlap of DE genes (Fig. 6f). In conclusion, Normalisr reconstructed specific and robust gene regulatory networks from high-MOI CRISPRi screens that could be cross-validated with existing bulk datasets and domain knowledge.

**Normalisr discovered functional gene modules with single-cell co-expression network in dysfunctional T cells.** Organisms are evolved to efficiently modulate various functional pathways, partly through the regulation and co-regulation of gene expression[17]. Manifested at the mRNA level, gene co-expression may provide unique insights into gene and cell functions. However, no existing co-expression detection or network inference method could control for false discovery at the single-cell, single-time-point level[5]. As shown with synthetic null datasets, Normalisr can account for nonlinear confounding from library size and therefore control the FDR in co-expression detection.

We used Normalisr to infer single-cell transcriptome-wide co-expression networks in dysfunctional CD8+ T cells in human melanoma MARS-seq[35]. After removal of low-variance outlier samples (9 of 15,537, Fig. S12), co-expression predominantly arose from cell-to-cell variations in housekeeping functions (Fig. S13 and "Methods"). The top 100 principal genes—those with the most co-expression edges—were enriched with cytosolic and ribosomal genes and reflected translational activity differences across cells (Fig. 7a and Supplementary Data 5). These housekeeping pathways dominated the correlations in mRNA expression and obscured the cell-type-specific pathways and interpretations of the co-expression network. Meanwhile, spike-ins clustered strongly together and with pseudogenes despite the unbiased analysis, suggesting limitations of spike-ins as a gold standard for null co-expression.

To focus on cell-type-specific pathways, we developed a subroutine to remove the high confidence but generic co-expression networks as additional covariates ("Methods"). For this, we identified the strongest Gene Ontology (GO) enrichment from the top 100 principal genes. We then introduced the top PC of expression of the genes in this GO category as an additional covariate before re-computing the co-expression statistics and the GO enrichment of top principal genes. Iteratively, we removed the top PCs of GO processes corresponding to "cytosolic part" (reflecting translation) and "chromosome condensation" (reflecting cell cycle). The top GO enrichments subsequently reflected immune system processes, signifying recovery of cell-type-specific co-expression networks and also the end of iteration (Figs. 7b and S14, and Supplementary Data 5).

Meanwhile, we performed single-cell DE between dysfunctional and naive T cells and recapitulated upregulation of major co-receptor genes associated with T cell dysfunction[47]—*TIGIT*, *HAVCR2/TIM-3*, *PDCD1*, and *LAG3* (Fig. S15 and Supplementary Data 6). Known gene sets in T cell dysfunction[47–49] were significantly enriched in our upregulated vs. downregulated genes. Normalisr recovered biologically relevant and validated DE from human melanoma MARS-seq data.

We integrated the single-cell DE analysis onto the co-expression network of dysfunctional T cells to understand gene co-expression and clustering patterns from cell-to-cell variations

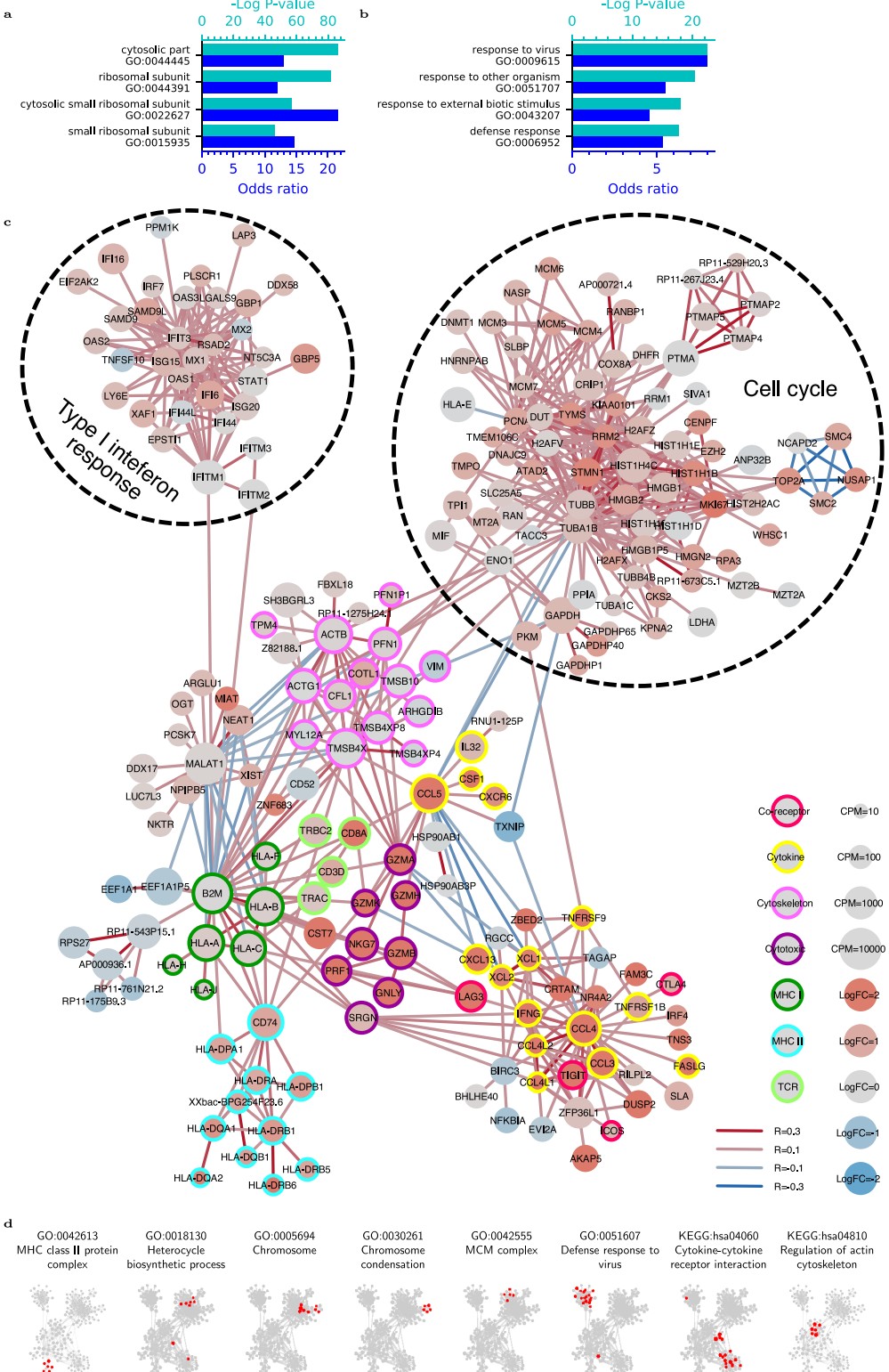

**Figure 7 Normalisr revealed gene-level cellular pathways and functional modules in the single-cell co-expression network. a**, **b** Single-cell co-expression was dominated by house-keeping programs (**a**), whose iterative removal recovered cell-type-specific programs (**b**), according to the top 4 GO enrichments (*Y*) of the top 100 principal genes in the co-expression network. *P* values and odds ratios are shown in cyan (top *X*) and blue (bottom *X*). **c** Single-cell transcriptome-wide co-expression network (major component) after house-keeping program removal highlighted functional gene sets for dysfunctional T cells. Edge color indicates positive (red) or negative (blue) co-expression in Pearson *R*. Node color indicates DE logFC between dysfunctional and naive T cells. Node size indicates average expression level in dysfunctional T cells. Node border annotates known gene functions. Dashed circles indicate major gene co-expression clusters. **d** Single-cell co-expression recovered cell-type-specific and generic gene functional similarities, according to significant over-abundances of edges between genes in the same GO or KEGG pathway, as highlighted in red.

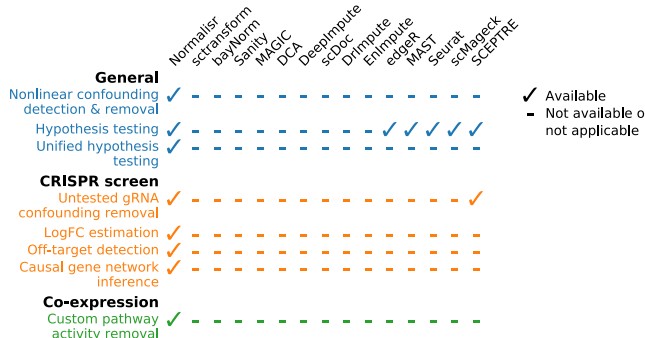

**Figure 8 A list of Normalisr's featured functions and their availabilities as reported in other single-cell methods.** Normalisr features several unique functionalities for general-purpose normalization and specifically for single-cell CRISPR screen or gene co-expression analyses.

(Fig. 7c and Supplementary Data 7). We observed two distinct gene clusters of cell cycle (whose secondary PCs remained evident in co-expression despite the removal of its top PC) and type I interferon response (Supplementary Data 8). We annotated the remaining, interconnected genes according to their known roles in T cell function in knowledge-base[50] and literature into one of the seven categories: cytokines (and receptors), cytoskeleton, cytotoxic, major histocompatibility complex (MHC) class I, MHC class II, T cell receptors (TCRs), and co-receptors. Genes in the same functional category formed obvious regional co-expression clusters. Between the clusters, MHC class I genes neighbored MHC class II and TCR genes, whereas effector genes, including co-receptor, cytokine, and cytotoxic genes, were more closely linked and collectively upregulated. *CCL5*, *CSF1*, *CXCR6*, and *IL32* formed a separate co-expression cluster that is negatively associated with the rest of the cytokine program. Expression of dysfunctional genes, including *CTLA4*, *LAG3*, and *TIGIT*, were also correlated with cytokine activity. This cluster comprised a mixture of immune activation and exhaustion genes as well as sub-clusters divided by negative correlations. The co-expression network recovered by Normalisr suggests potential functional diversification in the dysfunctional T cell population, in agreement with previous discoveries in this field[51].

We then evaluated the functional associations recoverable from single-cell co-expression based on the over-abundance of co-expression edges between genes in each GO or Kyoto Encyclopedia of Genes and Genomes (KEGG) annotation. Despite the previously reported difficulty in recapitulating GO annotations from single-cell co-expression networks from over 1000 cells[52], we found that genes in 33 GO and 8 KEGG pathways had significantly more co-expression networks than randomly assigned annotations (Bonferroni $P \leq 0.05$, Supplementary Data 9). These pathways encompassed a wide range of cell-type-specific and generic functions (Fig. 7d and Supplementary Data 7). Overall, this analysis demonstrated that Normalisr recovered gene-level cellular pathways and functional modules in high-quality single-cell transcriptome-wide co-expression networks.

## Discussion

The current rise in scRNA-seq data generation represents a tremendous opportunity to understand gene regulation at the single-cell level, such as through pooled screens or co-expression. Here we describe Normalisr as a unified normalization–association two-step inferential framework across multiple experimental conditions and scRNA-seq protocols with several unique or rare functionalities (Fig. 8). Normalisr efficiently infers gene regulatory networks from pooled single-cell CRISPRi screens and

reduces false positives from gRNA cross-associations or off-target effects. Normalisr removes house-keeping modes in scRNA-seq data and infers transcriptome-wide, FDR-controlled co-expression networks consisting of cell-type-specific functional modules.

Normalisr addresses the sparsity and technical variation challenges of scRNA-seq with posterior mRNA abundances, non-linear cellular summary covariates, and variance normalization. It fits in the framework of linear models[53] and achieves high performance over a diverse range of frequentist inference scenarios. Normalisr enables high-quality gene regulation and co-regulation analyses at the single-cell, single-gene level, and at scale. However, Normalisr is not designed for hypothesis testing of arbitrary nonlinear associations[54], reconstructing causal regulatory networks through v-structures or regularized regression[55], or searching for Granger causality with longitudinal information[4]. Integrative studies spanning multiple datasets may also require additional consideration for batch effects[56]. We only focused on UMI-based datasets that are typically easier to scale to many (≥10,000) cells, e.g., with mature commercial solutions and limited adjustments, which is a key advantage for studies like single-cell CRISPR screens and cell-type-specific co-expression networks. Full-length-based platforms may produce a different distribution for read counts[57], which may benefit from revalidating the nonlinear cellular summary covariates in Normalisr (Fig. S3).

We established a restricted forward stepwise selection process to systematically dissect nonlinear effects of known confounders at a reduced burden of sensitivity loss or overfitting. This offers an unbiased and generalizable approach to reduce problem-specific designs in normalization or association testing methods. We avoided informative prior or imputation to maintain independence between genes or cells for statistical inference but nevertheless could outperform methods based on such techniques. Consequently, the unbiasedly determined nonlinear covariates were consistent across different scRNA-seq protocols such as 10× Genomics, MARS-seq, and CROP-seq at different sequencing depths.

Linear models possess immense capacities and flexibilities for scRNA-seq, such as "soft" groupings for DE and kinship-aware population studies. At the same sequencing depth as bulk RNA-seq, scRNA-seq additionally partitions the reads between cells and cell types. This extra information provides substantial statistical gain and cell-type stratification.

## Methods

**MMSE estimator of log gene expression.** Normalisr regards sequencing as a binomial sampling in the pool of mRNAs, i.e. the read/UMI count matrix $g_{ij} \mid n_j^{(g)}, \tilde{g}_{ij} \sim B(n_j^{(g)}, \tilde{g}_{ij}) \in \mathbb{N}^0$ for gene $i = 1, \ldots, n_g$ in cell $j = 1, \ldots, n_c$, where $n_j^{(g)} \equiv \sum_k g_{kj}$ is the empirical sequenced mRNA count for cell $j$ and $\tilde{g}_{ij}$ is the biological proportion of mRNAs from gene $i$ in cell $j$. With a non-informative prior (standard uniform distribution), the posterior distribution is $\tilde{g}_{ij} \mid g_{ij} \sim Beta(1 + g_{ij}, 1 + n_j^{(g)} - g_{ij})$. The MMSE estimator for log relative gene expression (here named Bayesian log expression) is $y_{ij} \equiv \mathbb{E} \ln \tilde{g}_{ij} = \psi(1 + g_{ij}) - \psi(2 + n_j^{(g)})$, where $\psi$ is the digamma function (Supplementary Information). Natural log is used throughout this paper.

**Cellular summary covariates.** To minimize spurious, technical gene inter-dependencies that may interfere with true co-expression patterns, we hypothesized and restricted covariate candidates within cellular summary statistics, including log total read count and the number of 0-read genes, defined as $\mathbf{C}^{(\mathrm{orig})}$ here. To account for their potential nonlinear confounding effects, we used restricted forward stepwise selection with the optimization goal of uniformly distributed (linear) co-expression $P$ values on null datasets, i.e., simulated read count matrices based on real datasets but without any underlying co-expression.

Specifically, consider gene expression matrix $\mathbf{G} \in \mathbb{R}^{n_g \times n_c}$ as Bayesian log expression, and the original cellular summary covariates $\mathbf{C}^{(\mathrm{orig})} \in \mathbb{R}^{n_{\mathrm{cov}} \times n_c}$ where $n_{\mathrm{cov}}$ is the number of the original cellular summary covariates. Assume that $\mathbf{G}$ is linearly confounded at mean and variance levels by unknown, fixed, nonlinear column-wise functions $\mathbf{C}(\mathbf{C}^{(\mathrm{orig})}) \equiv (\mathbf{C}_1(\mathbf{C}^{(\mathrm{orig})}), \mathbf{C}_2(\mathbf{C}^{(\mathrm{orig})}), \ldots)^T \in \mathbb{R}^{\cdot \times n_c}$, leading to false positives of gene co-expression even under the null hypothesis. We

performed a Taylor expansion of each $\mathbf{C}_i(\mathbf{C}^{(orig)})$, the nonlinear confounder function $i$, as

$$
\mathbf{C}_i(\mathbf{C}^{(orig)}) = \sum_{\substack{\mathbf{p}\,\in\,(\mathbb{N}^0)^{n_{cov}} \\ (\sum_{k=1}^{n_{cov}} p_k)\,\le\,n}} \alpha_i^{(\mathbf{p})} \prod_{k=1}^{n_{cov}} \left(\mathbf{C}_k^{(orig)}\right)^{p_k} + \sum_{\substack{\mathbf{p}\,\in\,(\mathbb{N}^0)^{n_{cov}} \\ (\sum_{k=1}^{n_{cov}} p_k)\,=\,n+1}} O\left(\prod_{k=1}^{n_{cov}} \left(\mathbf{C}_k^{(orig)}\right)^{p_k}\right),
$$
(1)

where operations are element-wise (i.e. cell-wise). The first term represents multivariate Taylor expansion up to total order $n$ whose coefficients are defined as $\alpha_i^{(\mathbf{p})}$, and the second term contains higher orders. Therefore, nonlinear confounders $\mathbf{C}$ can be accounted for altogether with higher-order sums of the original covariates $\mathbf{C}^{(orig)}$, provided the series converge quickly (subjecting to validation for the data type).

Therefore, we can redefine $\mathbf{C}_i$ as each of the polynomial terms in the first term of Eq. (1)'s r.h.s. to determine the important nonlinear covariates. We iteratively included nonlinear covariates (parameterized by $\{\mathbf{p}|\sum_i \alpha_i^{\mathbf{p}} \ne 0\}$). Starting from the constant intercept covariate set $\mathcal{C} = \{1\}$, in each step we introduced the optimal candidate covariate to improve null co-expression $P$ value distribution toward uniform distribution from the set

$$
\left\{ \prod_{k=1}^{n_{cov}} \left(\mathbf{C}_k^{(orig)}\right)^{p_k} \,\middle|\, \forall j \in \{l|p_l > 0\}, \left(\mathbf{C}_j^{(orig)}\right)^{-1} \prod_{k=1}^{n_{cov}} \left(\mathbf{C}_k^{(orig)}\right)^{p_k} \in \mathcal{C} \right\}.
$$
(2)

This restricted the candidate nonlinear covariate set to series expansion terms whose all lower-order terms had already been included. The restriction reduces the number of models/covariate candidates to a finite (and small) number, while only assuming that the aggregated effect of all $\mathbf{C}_i$ functions is generically nonlinear, i.e., not very close to even or odd. The iteration ends when none could provide obvious improvement, and then the nonlinear covariate terms $\mathbf{C}_i$ (each defined by the polynomial order vector $\mathbf{p}$) can be recovered from the covariate set $\mathcal{C}$. The restricted forward stepwise selection avoids unnecessary covariates, which degrade statistical power and reduce generalizability.

For Normalisr, the forward selection goal was the uniform distribution of null co-expression $P$ values. The forward selection process introduced three covariates: log total read count, its square, and the number of 0-read genes.

**Existing covariate aggregation.** Existing covariates ($\mathbf{C}^{(e)}$) can contain batches, pathways that are not the analytical focus (see GO pathway removal in co-expression networks), constant intercept term, etc. Aggregation of existing and cellular summary covariates is a simple concatenation

$$
\mathbf{C}^{(a)} = \left(\mathbf{C}^{(e)T}, \mathbf{C}^T\right)^T,
$$
(3)

followed by an orthonormal transformation (excluding intercept term).

**Cell variance estimation.** Covariates may confound (Bayesian log) expression at the variance level in addition to the mean, e.g., due to the sparse multinomial sampling in sequencing. To account for such effects and to prepare for their removal in the next step, we estimated the contribution to unexplained expression variance from covariates. For this, we first modeled $\mathbf{Y} = \alpha^{(y)}\mathbf{C}^{(a)} + \varepsilon^{(y)}$ in matrix form for confounding at mean level, where $\alpha^{(y)} \equiv \{\alpha_{ij}^{(y)}\}$ is the matrix of covariate $j$'s effect on gene $i$'s mean expression and is unbiasedly estimated with the maximum likelihood estimator (MLE) from linear regression as $\hat{\alpha}^{(y)} = \mathbf{Y}\,\mathbf{C}^{(a)T}(\mathbf{C}^{(a)}\mathbf{C}^{(a)T})^{-1}$. To compute and model the unexplained variance, as oppose to independent consideration of each gene $i$ with $(\varepsilon_i^{(y)})^2$, we fit a general variance confounding model for all genes with their combined error variance $v_k \equiv \sum_i (\varepsilon_{ik}^{(y)})^2$ for each cell $k$ with $\ln \mathbf{v} = \alpha^{(v)}\mathbf{C}^{(a)} + \varepsilon^{(v)}$ with $\varepsilon^{(v)} \sim i.i.d\ N(0,\sigma^2)$. This reduces the number of intermediate variables and consequently overfitting. Then the MLE of the estimated cell variance is $\hat{\mathbf{v}} = e^{\hat{\alpha}^{(v)}\mathbf{C}^{(a)}}$ where $\hat{\alpha}^{(v)} = \ln \mathbf{v}\,\mathbf{C}^{(a)T}(\mathbf{C}^{(a)}\mathbf{C}^{(a)T})^{-1}$. Note that maximum likelihood optimization with $\hat{\alpha}^{(y)}$ and $\hat{\alpha}^{(v)}$ together fails due to overfitting and prioritization of few cells. This regression-based cell variance estimation retains changes of expression variance that are due to biological sources.

**Gene expression variance normalization.** Gene expression levels were transformed to $\tilde{y}_{ij} = \hat{y}_{ij} + (y_{ij} - \hat{y}_{ij})/\hat{v}_i^{\gamma_i/2}$ to normalize variance in the second term, where $\hat{\mathbf{Y}} = \hat{\alpha}^{(y)}\mathbf{C}^{(a)}$. The scaling factor $\gamma_i \equiv \bar{y}_i / \max_k \bar{y}_k$, with $\bar{y}_k \equiv \#_j(g_{kj} = 0)$, smoothly scales variance normalization effect to zero on highly expressed genes as their expressions are already accurately measured in scRNA-seq. Covariates' variances were also scaled in full to $\tilde{\mathbf{C}}_{ij} = \mathbf{C}_{ij}^{(a)}/\hat{v}_j$ accordingly with the exception of categorical covariates left unchanged in one-hot encoding.

**Gene differential and co-expression hypothesis tests.** Using normalized expression and covariates, DE was tested with linear model $\tilde{\mathbf{Y}} = \alpha\mathbf{s} + \beta\tilde{\mathbf{C}} + \varepsilon$,

where $s = 0, 1$ indicates cell set membership (case vs. control) and $\varepsilon \sim i.i.d\ N(0,\sigma^2)$. LogFC was estimated as $\hat{\alpha}$ using maximum likelihood. The two-sided $P$ value was computed for the null hypothesis $\alpha = 0$ using the exact null distribution of the proportion of explained variance (by $\mathbf{s}$), as $\text{Beta}(1/2, (n_c - 1 - \text{rank}(\tilde{\mathbf{C}}))/2)$. Co-expression between genes $i, j$ was tested with $\tilde{\mathbf{Y}}_i = \alpha\tilde{\mathbf{Y}}_j + \beta\tilde{\mathbf{C}} + \varepsilon$, with the same setup otherwise. Their (conditional) Pearson correlation and $P$ value were computed from $\alpha$ and are symmetric between $i$ and $j$.

**Outlier cell removal.** For outlier removal, inverse of (estimated) cell variance was modeled with normal distribution. Given the prior bound of outlier proportion as $r$, the iterative outlier detection method started with $r$ to $1 - r$ percentiles as non-outlier cells. In each iteration, a normal distribution was fitted with MLE for the inverse variances of non-outliers, and was used for outlier test of all cells with two-sided $P$ values. Cells below the given Bonferroni adjusted $P$ value threshold were regarded as outliers in the next iteration. This was repeated until convergence and the prior bound of outlier proportion $r$ was checked. The removal process takes the prior bound of outlier proportion $r$ and the $P$ value threshold as parameters.

**GO pathway removal in co-expression networks.** The GO pathway removal took an iterative process given the significance cutoff for co-expression Q-value. In each iteration, Normalisr first identified the top 100 principal genes in the co-expression network, defined as those with most co-expressed genes (passing the significance threshold). To find the dominating pathway for co-expression, Normalisr performed GO enrichment analysis on these regulators using all genes after QC as background. The most enriched GO term (by $P$ value) was manually examined for contextual relevance and house-keeping role. The researcher then chose to proceed the removal or stop iteration.

If removal was elected, Normalisr first removed existing covariates linearly from the (Normalisr normalized) transcriptomic profile. Then Normalisr extracted the top PC of the subset transcriptomic profile of genes in the GO term. This PC was added as an extra covariate, before producing a new co-expression network and repeating the iteration.

This process was performed iteratively until the most enriched GO term was contextually relevant, as judged by the researcher. Here we stop the iteration when the GO term is specific to the cell type (dys)function (immune function for dysfunctional T cells). The only manual input of the iterative process was when to stop iteration.

**Random partition for null DE.** Homogenous cells from the same dataset, cell type, and condition were partitioned to two sets randomly for 100 times for null DE. Each random partition had a different partition rate, sampled uniformly between 5 and 50%.

**Existing normalization and imputation methods.** We used the default configurations in DrImpute (1.0), EnImpute (1.1), scDoc (0.0.0.9), DeepImpute (1.2), sctransform (variance stabilizing transformation, 0.2.1), bayNorm (separately with mode_version or mean_version, 1.2.0), Sanity (1.0), DCA (0.2.3), and MAGIC (1.2.1), with 50 latent dimensions for MAGIC. Their differential and co-expression used the same linear model but without cellular summary covariates. Log(CPM + 1) transformation was applied on DeepImpute and DrImpute outputs because they still contained zeros. EnImpute could not connect with scImpute, Seurat, and ALRA and was based only on the remaining methods. Sctransform's log10 expression was converted to natural log for logFC comparison. ScImpute[58] and ZINB-WaVE[59] were not included because they could not accommodate the data size with excessive memory and/or time requirements. ScVI[60] could not fit into our comparison because it did not recommend $P$ value-based FDR estimation. Existing methods were evaluated on their intended usage in publications and/or tutorial because their combinatorial uses are exponentially many but the statistical motivation and assumption behind the combination is often less clear and less scrutinized.

**Existing DE methods.** For edgeR and MAST, we exactly replicated the codes in ref. [8], using edgeRQLFDetRate for edgeR (3.26.8)[9] and MASTcpmDetRate for MAST (1.10.0)[1]. DE with Seurat (4.0.3) included its normalization and DE functions (NormalizeData, ScaleData, and FindMarkers with the default Wilcoxon test) from raw read counts according to its tutorial. Log2 fold change from edgeR and Seurat was converted to natural log for comparison.

**Evaluation of null $P$ value distribution in hypothesis testing.** For differential gene expression, we split genes to ten equally sized groups by the number of expressed cells, low to high. We then evaluated the null distribution of $P$ values for each group separately. This can reveal null $P$ value biases that depend on the expression level. For the same purpose, we evaluated the null distribution of gene co-expression $P$ values for each pair of gene groups separately.

Two-sided KS test was performed on each of the 10 groups of null $P$ values for DE, or 55 groups for co-expression, to evaluate how they represent a standard uniform distribution. The resulting 10 or 55 KS test $P$ values were drawn and

compared across different methods with violin plot. Larger KS test $P$ values indicate better recovery of uniform null $P$ values.

**Running time evaluation**. Each method was timed after QC on a high-performance computer of 256 CPU cores. Initial and final disk read/write was excluded for timing for methods natively in Python.

**Synthetic null co-expression dataset of homogeneous cells**. To produce a synthetic read count matrix $\tilde{G} = \{\tilde{g}_{ij}\}$ without co-expression of $\tilde{n}_c$ cells and $\tilde{n}_g$ genes, yet mimicking a real, pre-QC dataset with read count matrix $G = \{g_{ij}\}$, of $n_c$ cells and $n_g$ genes, we took the following steps.

1. Compute empirical distributions of read counts per cell ($\sum_i g_{ij}$) as $\mathcal{D}_c$ and total read proportions per gene ($\sum_j g_{ij}/\sum_{ij} g_{ij}$) as $\mathcal{D}_g$.
2. $n_j$: Sample/simulate each cell $j$'s total read count from $\mathcal{D}_c$ and scale them by $\tilde{n}_g/n_g$.
3. $p_i$: Sample each gene $i$'s mean read proportion from $\mathcal{D}_g$.
4. $b_{ij} \sim i.i.d\ N(0, \sigma^2)$: Simulate biological variations.
5. $\tilde{p}_{ij} = p_i e^{b_{ij}}$: Compute expression proportion.
6. $\tilde{g}_{ij}$: For cell $j$, sample $n_j$ reads for all genes, with weight $\tilde{p}_{ij}$ for gene $i$.

The simulation contains three hyperparameters: the number of cells $\tilde{n}_c$, the number of genes $\tilde{n}_g$, and biological variation strength $\sigma$. The simulation model is parameterized by the distributions of read counts per cell and read proportions per gene, which are estimated with the empirical distributions $\mathcal{D}_c$ and $\mathcal{D}_g$ in step 1. This simulation reflects the confounding effect of sequencing depth in real datasets on a homogeneous cell population, which is lost by permutation-based null datasets. This model is similar with powsimR's negative binomial model[61], except that it separates biological variations from the binomial detection process, allows for extra method evaluations with the ground-truth of biological mRNA proportions $\tilde{p}_{ij}$, and accepts biological variation strength $\sigma$ as input. The ground-truth logFC for gene $i$ based on the simulation data is defined as the difference in mean log expression proportion, i.e., $\frac{1}{\sum_j m_j}\sum_j m_j \ln \frac{\tilde{p}_{ij}}{\sum_k \tilde{p}_{kj}} - \frac{1}{\sum_j 1 - m_j}\sum_j (1 - m_j)\ln \frac{\tilde{p}_{ij}}{\sum_k \tilde{p}_{kj}}$ where $m_j = 0, 1$ indicates cell $j$'s binary membership of two cell subsets to be compared.

**Synthetic datasets from SymSim and Splatter**. We used SymSim (0.0.0.9) and Splatter (1.14.1) for simulating datasets with known logFC in DE. The parameters were first estimated with BestMatchParams (for SymSim) or splatEstimate (for Splatter) from a given dataset (Perturb-seq or MARS-seq). The estimated parameters were then used to simulate a dataset of given gene and cell counts following their tutorial for DE datasets. Their internal ground-truth logFCs were stored besides the simulated raw count matrix to benchmark logFC estimation by different methods. The numbers of cells in the minor group for simulating DE were generated with random keystrokes as 519, 1134, 2656, and 3927 in totally 8472 cells.

**DE logFC estimation evaluations**. Estimated logFCs were compared against the known ground-truth in synthetic datasets for method evaluation. To decompose estimation errors into bias and variance, we trained a univariate, intercept-free linear model to predict estimated logFCs with the ground-truth logFCs. The resulting regression coefficient represents the overall bias in logFC scale estimation. The goodness of fit (Pearson $R^2$) indicates the variance of logFC estimation. This evaluation was performed on separate gene groups from low to high expression levels (in terms of proportion of expressed cells) to better characterize method performances. The ideal method should provide low bias, low variance, and stable bias that changes minimally with expression level.

**Reproducibility across scRNA-seq protocols and experimental conditions**. We used the MARS-seq dataset[35] of human melanoma tissue samples at a lower sequencing depth to evaluate Normalisr's performances across different scRNA-seq protocols and experimental conditions. Only annotated non-live, dysfunctional T cells from frozen samples were selected as a relatively homogeneous population for method evaluation. All compatible evaluations were repeated on the MARS-seq datasets.

**Technical bias removal effects on cell population structure on UMAP embeddings**. We reduced the normalized or imputed transcriptome matrix to the top 50 PCs with PC analysis and then to the top two UMAP dimensions. Covariants were removed at mean level before dimension reduction, if present.

**ScRNA-seq QC**. In this study, we restricted our analyses to cells with at least 500 reads and 100 genes with non-zero reads, and genes with non-zero reads both in at least 50 cells and in at least 2% of all cells for 10× technologies, or in at least 500 cells and in at least 2% of all cells for MARS-seq, due to their different read count distributions in this study. The QC was performed iteratively until no gene or cell is

removed. Simulated datasets followed the criteria of the corresponding real dataset technology.

**FDR control and estimations of FPR and Q-value**. FPR was estimated with $\hat{\pi}_1$ in Storey's method[37] using `fdrtools`[62]. For CRISPRi analyses, this was performed separately for each gRNA–gene pair type.

True FDR is unknown without ground-truth but can be estimated and statistically controlled. For the purpose of FDR control, we used Q-values estimated with Benjamini–Hochberg procedure. To account for variations in gRNA specificity and gene role in CRISPRi analyses, Q-values were estimated for all gRNA–gene pairs in the "TSS, target" type together and separately for each gRNA in other types. We also used other statistical measures such as Bonferroni-adjusted $P$ values at different stringency levels when the goal was not controlling the FDR in a single set of exact hypothesis tests.

**Gene Ontology**. GO enrichment with `GOATOOLS` (0.8.4)[63] was restricted to post-QC genes as background, except in co-expression network modules where all known genes were used. To avoid evaluation biases for our expression-based analyses, for all GO analyses we excluded GO evidences that may also have an expression-based origin (IEP, HEP, RCA, TAS, NAS, IC, ND, and IEA). All three GO categories were used. GO enrichment $P$ values are raw unless stated otherwise. Gene names were converted with `mygene` when needed.

**High-MOI single-cell CRISPRi screen dataset**. UMI read count and gRNA assignment matrices were downloaded from Gene Expression Omnibus (GEO) and used for all analyses. Meta-data of gRNAs were downloaded from the Supplementary Materials of ref. [7].

**gRNA cross-association test**. Odds ratio of gRNA intersection was computed for every gRNA pair, with one-sided hypergeometric $P$ values.

**Types of gRNA–gene pairs**.

- Naive/aware, target: NTC gRNA vs. genes targeted by any gRNA at its TSS
- Naive/aware, other: NTC gRNA vs. genes not targeted by any gRNA at its TSS
- TSS, target: TSS-targeting gRNA vs. the gene it targets
- TSS, other: TSS-targeting gRNA vs. genes it doesn't target
- NTC: NTC gRNA vs. any gene
- Enhancer: Enhancer-candidate-targeting gRNA vs. any gene

**Association $P$ value and logFC histograms in single-cell CRISPR screen analysis**. Histograms were constructed with 50 bins, while being symmetric for logFC. Separate histograms were drawn for different gRNA–gene pair types. Absolute histogram errors were estimated as $2\sqrt{N+1}$, where $N$ is the number of occurrences at each bin.

**Competition-aware method in single-cell CRISPR screen analysis**. The competition-aware method regards all other, untested gRNAs as covariates. For efficiency, these covariates were introduced at the time of association testing and were assumed to only affect the mean expression, and all covariates were reduced to top 500 PCs if >10,000. Other numbers of top PCs have been tried and found reliable against no dimension reduction on the small-scale dataset.

**Comparison with existing single-cell CRISPR screen analysis methods**. We downloaded the author's deposition of SCEPTRE's gene-level $P$ values on the same full-scale K562 study for comparison of sensitivity on the repression effects of TSS-targeting gRNA as positive controls. To enable comparison with SCEPTRE, gRNA level $P$ values from Normalisr were combined to gene level with Fisher's method. $P$ value strength comparison was limited to positive control genes that were highly significant according to either Normalisr or SCEPTRE (Bonferroni $P < 0.05$). $P$ values for non-cis-effects (including from NTCs) were not available from SCEPTRE for sensitivity or specificity comparison.

ScMAGECK was not compared for sensitivity because it did not finish computation within 2 weeks and scMAGECK's $P$ values were not publicly available for the same study. Running time projections were based on ref. [13] for SCEPTRE and Fig. 2c for Seurat, edgeR, and MAST.

**CRISPRi off-target rate**. For each gRNA-targeted gene, a weaker and a stronger gRNA targeted its TSS, according to the $P$ value of their linear associations. Trans-genes that associated significantly ($Q \le 0.05$ or $10^{-5}$ as specified in figure legend) with the weaker gRNA but highly insignificantly ($P \ge 0.1$) with the stronger gRNA were regarded as off-targets. The raw off-target rate for each weaker gRNA was defined as the proportion of off-targets among significant trans-targets of the weaker gRNA (under the same $Q$ threshold). It is also the FDR when evaluating the weaker gRNA's associations using the stronger one as gold standard. The estimated

off-target rate for each weaker gRNA was defined as min(raw off-target rate/(1 − 0.1), 1) to extrapolate and account for the proportion of insignificant associations with the stronger gRNA that was lost in choosing $P \geq 0.1$. The (overall) off-target rate for a screen was defined as the average estimated off-target rate across the weaker gRNAs.

**Inferring gene regulations from *trans*-associations in single-cell CRISPR screen.** Inferred gene regulations must satisfy all the following conditions for both gRNAs in both screens.

- Significant repression of targeted gene: $Q \leq 0.05$ and logFC $\leq -0.2$.
- Significant association with *trans*-gene: $Q \leq 0.2$, $\left|\text{logFC}\right| \geq 0.05$, relative (to logFC of targeted gene) $\left|\text{logFC}\right| \geq 0.05$.
- No other gene repressed within 1 Mbp up- and down-stream window of targeted TSS: $P \leq 0.001$ and logFC $\leq -0.1$. This aims to avoid identifying the scenario: intended target ← gRNA → *cis*-gene → another gene, which would otherwise become a false positive of gene regulation: gRNA → intended target → another gene.

**Dysfunctional T cells in human melanoma dataset.** We downloaded the read count matrix and meta-data from GEO, after QC from the original authors. We regarded the following as categorical covariates that may confound gene (co-) expression: batches from amplification, plate, and sequencing, as well as variations from patient, cell alive/dead, sample location, and sampling processing (fresh or frozen). In our QC, cells from donors that had <5 cells of the same cell type were discarded. With the prior bound for outlier proportion as 2%, low-variance outlier cells with Bonferroni $P \leq 10^{-10}$ were removed from downstream analyses with Normalisr. Pseudo-genes and spike-ins were treated indifferently in statistical inference.

**Dysfunctional gene overlap testing.** Overlap testing was performed between known dysfunctional genes (from literature) and genes upregulated ($Q \leq 0.05$) in dysfunctional vs. naive T cells. Background gene set of this dataset was selected as upregulated or downregulated genes ($Q \leq 0.05$). Only known dysfunctional genes found in the background set were considered for hypergeometric testing $P$ value.

**Gene co-expression network in dysfunctional T cells.** We computed $Q$-value networks from raw $P$ values separately for each gene against all other genes, to account for different gene roles such as principal genes or master regulators[22]. For co-expression network in dysfunctional T cells, we used a strong cutoff ($Q \leq 10^{-15}$) to prioritize more direct gene interactions. The final co-expression network focused on the major connected component of the full co-expression network after two iterations of GO pathway removal. Within the final network, we also removed a relatively separate cluster consisting solely of non-coding mRNAs and spike-ins (Fig. S14).

**Over-abundance of co-expression networks within annotation group.** We used GOATOOLS[63] and BioServices[64] for GO and KEGG pathways, respectively. We restricted the gene sets to having at least two edges but at most half in the given co-expression network. To reduce multiple testing, for each GO term, its parent term is excluded if it has the same annotation on the network. $P$ values for edge over-abundance were computed with random annotation assignment with the same number of annotated nodes in the network.

**Reporting summary.** Further information on research design is available in the Nature Research Reporting Summary linked to this article.

## Data availability
The Perturb-seq data used in this study are available in the Gene Expression Omnibus database under accession code GSM2406681. The MARS-seq data used in this study are available in the Gene Expression Omnibus database under accession code GSE123139. The CROP-seq data used in this study are available in the Gene Expression Omnibus database under accession code GSE120861. The shRNA DDX3X knock-down and control data used in this study are available in the Gene Expression Omnibus database under accession codes GSM2406681 and GSM2138876 or the ENCODE database under accession codes ENCSR000KYM and ENCSR913CAE. SCEPTRE results were downloaded from https://drive.google.com/drive/folders/1ynZRMvGtFxfBiD0zAcuIYjNeS8Jj4AP9 in ref. [13].

## Code availability
Normalisr is publicly available at https://github.com/lingfeiwang/normalisr[65].

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

## Acknowledgements

L.W. would like to thank Daniel Graham, Kirk Gosik, and Heping Xu for early discussions, Jacques Deguine for insights in T cell biology, Ramnik Xavier and Luca Pinello for funding the computation, and Jacques Deguine and Qian Qin for extensive feedback on the manuscript. L.W. would like to acknowledge the ENCODE Consortium and Brenton Graveley's Laboratory for generating the shRNA *DDX3X* knock-down datasets. L.W. personally appreciates Lynn Rekhi's administrative assistance during the COVID-19 outbreak.

## Competing interests
The author declares no competing interests.
