## [Peer Review File · Nature Communications]

Single-cell normalization and association testing unifying
CRISPR screen and gene co-expression analyses with NormaliserReviewers' Comments:

Reviewer #1:

Remarks to the Author:

The authors developed Normalisr, a framework consisting of two step/components, normalization and association testing, for single-cell differential expression, co-expression, and CRISPR screen analyses. This pipeline would be interesting and useful in the relevant research area.

Main concerns:

- 1) In the association study, the linear model is applied in both (two group) DE analysis and gene co-expression analysis. The big assumption is that the lcpm follows a normal distribution. A residual checking should be performed for both synthetic data and real data to make sure the assumption is satisfied.
- 2) The proposed method is compared to a few normalization methods and also imputation methods, parallelly. According to a few existing review literatures, both normalization and imputation should be conducted for single-cell sequencing count data. The order of them matters too. The performance of Normalisr should be compared to these combinations rather than only one step preprocessing.
- 3) Regarding DE analysis, more DE methods should be compared, rather than only EdgeR and MAST.
- 4) More recently developed imputation methods should be compared to, e.g., enimpute, drimpute, scDoc, deepimpute etc.
- 5) The way for generating "Synthetic dataset of null co-expression" is very limited. There should be way more than 3 parameters to simulate a single cell count dataset, e.g., the proportion of zeros, the fraction of DE genes, the level of DE etc. You should use simulators Splatter and SymSim for data simulation.
- 6) Why did they use MMSE? what is the true value in this formula? Give the details of derivative of formula in L295.
- 7) The two summary statistics are actually correlated!! (L63 and see method) How would this affect the linear model fitting?
- 8) In L347 "For outlier removal, inverse of (estimated) cell variance was modeled with normal distribution" please use simulation and real data to demonstrate that the assumption is valid.
- 9) In L 350-351, "Cells below the given Bonferroni adjusted P-value threshold were regarded as outliers in the next iteration". The number of cells is usually large, Bonferroni correction is too conservative. You need to use appropriate way for multiple tests comparison.
- 10) What forward stepwise selection method is used? The threshold or parameter setting?
- 11) L359, "The most enriched GO term (by P-value) was manually examined for contextual relevance and house-keeping role". Please provide more objective steps rather than manual examination for readers.
- 12) Sometime the authors used FDR, Q-value, or Bonferroni correction. Please explain clearly and be consistent.
- 13) Can the DE analysis be extended to a more general case, i.e., not only case-control?
- 14) It would be helpful for the readers if the authors can provide a (cluster) visualization on the normalized expression as an intermediate output from Normalisr, since the possible confounders/covariates are already taken into account.

Minor concerns:

In Figure 1 b) the color of read counts should be grey, same as the one in a).

In Figure 1 b) the color of normalized expression should be same as the one (normalized expression) in a)

L 477, " $(Q \leq 0.05 \text{ or } 10^{-5})$ " why?

L487, why did you use $P \leq 0.001$ but used Q-value for L484 and L485?

L488, the absolute value of logFC should be used.

Why is binomial approximate of multinomial? In line 57

Reviewer #2:

Remarks to the Author:

The author recognizes that the cell-to-cell technical variability and low read counts are two main challenges in analyzing single-cell RNA-seq data. To overcome this, they anticipate a potential unified two-steps that first normalizes data to rule out nonlinear confounding and then infers association between gene expressions at a single cell resolution using linear models. The author suggests that their method outperforms other existing methods of single-cell RNA-seq data analysis by accessing direct evaluation of P-values (from conditional Pearson correlation tests) and demonstrating extremely short computation time compared to alternative approaches. The author applies this method to identify differentially expressed genes and gene co-expression both from standard scRNA-seq datasets and from single-cell CRISPR experiments. This is an inferential method that takes a novel approach to identifying and correcting known and suspected confounder covariates prior to assessing statistical associations.

Overall, I find the method sound from a statistical perspective. However, the manuscript would benefit from simpler language and more layman explanations on some technical details and on the reasoning behind experiments shown. As is, I worry the manuscript is sub-setting itself to a more specialized audience. In addition, the manuscript would benefit from additional information regarding the details of the datasets used and reasoning behind choices made during the method development. Finally, I am not fully convinced this method outperforms existing methods and suggest a more thorough evaluation using standardized datasets and evaluation frameworks described in reference [8].

Major points:

1. I have the following questions about the approach, which would benefit from more explanation in the text:

a. Minimum mean squared error estimator is a Bayesian method that requires information about the prior to estimate the posterior distribution. Could the authors be explicit on the specific parameters tested/assumed? It would be interesting to know how well the Bayesian approach compares to deterministic methods such as uniformly minimum variance unbiased estimator or a minimum variance unbiased estimator directly?

b. Modeling nonlinear effects with a Taylor Expansion is appropriate, but it would be informative to know what other non-linear methods were tested in modeling? Maybe there could be a quantitative comparison of these methods?

c. 'In contrast, edgeR and MAST, the best single-cell DE methods benchmarked in [8], had expression-dependent, 0- or 1-biased null P-values, which would lead to high false positive and false negative rates in application'. While the statement of 0 or 1 biased null p-values leading to higher FPR and FNR is seemingly self-evident, the authors should show data to demonstrate this explicitly if they wish to make this statement.

2. The MARS-seq dataset is used to show generalizability of the method, and the author concludes from this analysis that the results are consistent across multiple scRNA-seq protocols. However, there is no description on how this protocol differs from the Perturb-seq protocol or how the data was processed to make it consistent with that from the Perturb-seq experiments. In general, I find the evaluations of the ability of different methods to call differentially expressed genes somewhat limited. The Conquer dataset described in reference [8] has been specifically established to compare scRNA-seq pipelines by providing a repository of consistently processed, analysis-ready scRNA-seq datasets. That work also provides a clear framework that can be used to compare the performance of scRNA-seq pipelines. The author should take advantage of both to benchmark Normaliser against other tools in terms of: the number of genes that are found to be differentially expressed, the FPR, TPR, and AUROC. The author should also evaluate the robustness of their method to sample size and to UMI

versus full-length datasets.

One of the motivations to develop this novel method is that "A comprehensive benchmarking found single-cell-specific attempts in DE could not outperform existing bulk methods such as edgeR [8]". However, it is not clear to me that Normalizr achieves the goal of outperforming edgeR, particularly as edgeR is used as the "ground-truth" in many of the analysis performed (see below).

Minor points:

Specific points about the datasets used/analysis performed that would benefit from clarification:

Line 81: "We downloaded a high multiplicity of infection (high-MOI) CRISPRi Perturb-seq dataset of K562 cells [6]" Which dataset is this? From the materials and methods of the paper you reference it seems all Perturb-seq experiments are performed with a low MOI. Specifically, authors from that paper list only 3 single-cell CRISPR experiments (and corresponding infection rates):

- "pilot experiment" - transduction efficiency of 20%–30%
- "UPR epistasis experiment" - transduction efficiencies of 5%–10%
- "UPR Perturb-seq experiment" - transduction efficiency of 15%

All of these correspond to low, not high MOIs. Additionally, can you provide in the materials and methods the links from which the datasets were downloaded? Are you starting your analysis from raw counts? Unmapped reads? In addition, writing a summary on how these datasets were generated and why you selected them to benchmark your method would help to give context to the analysis you are presenting, and not force readers to go to the original publication

Line 95: how many cells have no detectable levels of gRNA expression? What is the size of your random groups? In the same line: what is the size of your "10 equally sized" gene subsets? How do you define the expression level bins?

Line 115: how is the "ground-truth logFC" defined?

Line 127-129: how are you measuring sensitivity here? Are you taking single cells expressing each of the gRNAs and checking if the targeted gene is significantly downregulated? How does this look at the level of individual cells? Specifically, how often for each method do you find that the targeted gene is significantly downregulated? In what fraction of cells does each method 'fail' to identify your true-positive gene? In supplementary Figure 6 you compare edgeR with several other methods. It would be more important for the purpose of validating your method to compare Normalizr with all other methods.

Line 132-134: It is rather strange to me that edgeR logFCs are being used as the ground-truth for your analysis, particularly since in Figure 2b you showed that in the null synthetic dataset it did not return a uniform null distribution of pValues and suggested that this could contribute to high false positive and false negative rates. Can you please explain your reasoning further?

In addition, you write: "Normalizr also estimated logFCs more accurately than other normalization or imputation methods when using edgeR as a proxy for ground-truth, which performed best in the synthetic dataset" Where is this statement quantified?

Line 144-146: Your false positive analysis is based on the hypothesis that cells may receive more than one gRNA leading to "competition between gRNAs for sCas9 or limited read counts", what is the evidence that this is actually the case? In addition, how many cells do you find in this dataset where with more than 1 assigned gRNA? What is the average gRNA number per cell? How many of those have NTC gRNAs?

Line 156-157: "Notably, the FPR was significantly higher among genes whose transcription start sites (TSS) were targeted by another gRNA" Where is the data supporting this statement?

Line 173-174: please provide details about what the "small screen". How many gRNAs? How many targeted genes? Are gRNAs designed against the TSS? What is the difference between this screen and the previous one?

Line 186-187: can you expand further on the analysis you are doing here and why? Are you looking for downstream effects of knocking down a gene to try to understand some of its functions?

Line 273: "However, Normalizr is not designed for nonlinear associations". I understand the author's sentiment, but this is confusing wording as your method does deal with nonlinearities.

Reviewer #3:

None

Reviewer #4:

Remarks to the Author:

The author presents a unified framework for normalisation and statistical association testing for single-cell CRISPR screens. Compared to the existing literature, this method (Normalizr) contributes with the following features:

- 1) a normalization step that models nonlinear confounding effects on gene expression by iteratively introducing higher order covariates;
- 2) improved sensitivity, specificity, and efficiency in the normalisation framework;
- 3) an association step that utilizes linear models to unify case-control DE, single-cell CRISPR screen analysis as multivariate DE, and gene co-expression network inference;
- 4) quantification of off-target effects;
- 5) higher computational efficiency.

The author demonstrates applications of this method in two scenarios: gene regulation screening from pooled CRISPRi CROP-seq screens and the reconstruction of transcriptome-wide co-expression networks from conventional scRNA-seq.

The method was tested by performing a wide range of evaluations and compared to other existing, relevant methods.

I believe that the contributions mentioned above are a valuable addition to the existing literature.

The method is well documented on GitHub; I could easily install the package and run the two examples:

`normalizr/examples/GSE120861`

`normalizr/examples/GSE123139`

I report few comments / questions below.

Line 29: "A comprehensive benchmarking found single-cell-specific attempts in DE could not outperform existing bulk methods such as edgeR".

Line 132: "Normalisr also estimated logFCs more accurately than other normalization or imputation methods when using edgeR as a proxy for ground-truth, which performed best in the synthetic dataset".

Comment 1) The reader might wonder what is the point of developing a single-cell-specific method in DE if this person can just use a method for bulk. I believe that this issue should be discussed more clearly in the text.

Line 95: "We partitioned cells that did not detect any gRNA into two random groups 100 times to evaluate the null P-value distribution of single-cell DE".

Comment 2) Why didn't the author use cells assigned to non-targeting control guides rather than partitioning cells based on lack of detection?

Line 485: "Significant association with trans-gene: $Q \leq 0.2$, $|\logFC| \geq 0.05$ "

Comment 3) The chosen threshold is barely significant.

Comment 4) Supplementary Figure 7 is reported in 4 pages, Supplementary Figure 12 in 3 pages. I would split figures and legends into, possibly, a single page dedicated to each figure.

Comment 5) Maybe the author could summarise and highlight in a table / schematic the features that Normalisr has and that the other methods don't have.

I deeply appreciate all Reviewers' time and effort in providing the valuable evaluations and feedbacks for the improvement of this manuscript. Your assessment and comments are extremely helpful in pointing out the level of details needed from multiple perspectives, and in suggesting more comprehensive evaluations and analyses. Following Reviewers' comments, I have performed the suggested evaluations and analyses, and improved the details and motivations of the method and analyses. I have updated the manuscript accordingly with changes highlighted in blue. Please find my point-by-point response to Reviewers' comments in blue below.

REVIEWER COMMENTS

Reviewer #1 (Expertise: normalization of single cell data):

The authors developed Normalizr, a framework consisting of two step/components, normalization and association testing, for single-cell differential expression, co-expression, and CRISPR screen analyses. This pipeline would be interesting and useful in the relevant research area.

I would like to acknowledge Reviewer #1's positive comments and suggestions on the manuscript. Following Reviewer #1's suggestions, I have included all the requested methods in the benchmark. The conclusion of this paper held equally well because Normalizr remained one of top performers in all the benchmarks after adding these methods. I have also added extra clarifications and analyses following Reviewer #1's other concerns and suggestions.

Main concerns:

1) In the association study, the linear model is applied in both (two group) DE analysis and gene co-expression analysis. The big assumption is that the lcpm follows a normal distribution. A residual checking should be performed for both synthetic data and real data to make sure the assumption is satisfied.

I would like to thank Reviewer #1 for pointing out the underlying assumption in linear models. Indeed, Bayesian logCPM (now called Bayesian log expression following Reviewer #3's comments) follows a distribution with improved normality (particularly lower skewness) than the conventional defined logCPM for both Perturb-seq and MARS-seq datasets which are more realistic than synthetic datasets (Fig 1 below). Although it does not follow a perfect normal distribution, linear models are known to be robust against deviations from normality (e.g. Ref [23], added in Lns 43-44). Moreover, most concerns over the violation of normality focus on several major statistics, such as type I/II error rates and bias/variance in effect size (Ref [23]), all of which have already been shown to be accurately estimated in this paper (Fig 2, S2-S8). Therefore, deviations from normal distribution is not a concern in this study.

Figure 1: **Bayesian log expression recovers distributions of expression residuals that satisfy the normality assumption better than conventional logCPM.** The violin plots show the distributions of skewness and kurtosis of residuals over genes for each normalization method, on the Perturb-seq (a) and MARS-seq (b) datasets. Smaller values (towards $-\infty$) indicate better normality. Bars indicate medians.

2) The proposed method is compared to a few normalization methods and also imputation methods, parallelly. According to a few existing review literatures, both normalization and imputation should be conducted for single-cell sequencing count data. The order of them matters too. The performance of Normalizr should be compared to these combinations rather than only one step preprocessing.

I fully agree with Reviewer #1 that the preprocessing outcome depends on the normalization-imputation combination as well as their order (e.g. Hou et al, Genome Biology 2020). However, the combinatorial use of normalization and imputation methods typically deviate from their intended usage as proposed in publications. There are exponentially many combinations yet the statistical motivation and assumption behind the combination is often less clear and less scrutinized. To have a clearly defined scope of study, this manuscript strictly follows the usage tutorial of published methods. I have added sentences to clarify the choice of methods compared (Lns 443-445).

3) Regarding DE analysis, more DE methods should be compared, rather than only EdgeR and MAST. I appreciate Reviewer #1's suggestion in more comprehensive benchmarking. EdgeR and MAST were included because they performed best in the previous benchmark study (Ref [8]). Following Reviewer #1's suggestion, I have added Seurat's differential expression (using Wilcoxon test by default) due to its popularity. Normalizr consistently outperformed Seurat in the benchmark (Fig 2, S4-S6, S8).

4) More recently developed imputation methods should be compared to, e.g., enimpute, drimpute, scDoc, deepimpute etc.

I would like to thank Reviewer #1 for suggesting imputation methods. I have included all four imputation methods. Normalizr consistently outperformed these methods in the benchmark (Fig 2, S2, S4-S9).

5) The way for generating "Synthetic dataset of null co-expression" is very limited. There should be way more than 3 parameters to simulate a single cell count dataset, e.g., the proportion of zeros, the fraction of DE genes, the level of DE etc. You should use simulators Splatter and SymSim for data simulation.

I would like to thank Reviewer #1 for suggesting simulation programs. Synthetic datasets were used in two evaluations: i) type I error rate in null co-expression from multinomial sampling, ii) logFC estimation in differential expression. For i), neither Splatter nor SymSim was designed to simulate co-expression free multinomial sampling. This simulation is a simplification of real scenarios but could recapitulate technical confounders and benchmark methods, as clarified in Lns 99-102. For ii), both Splatter and SymSim were added. Normalizr consistently remained as one of the top performers (Fig S5, S8, Lns 133-135, 139-141).

6) Why did they use MMSE? what is the true value in this formula? Give the details of derivative of formula in L295.

I wish to thank Reviewer #1 for suggesting more motivation and details. As inspired by Reviewer #2's question, the uniformly minimum variance unbiased estimator for log expression does not exist, which motivates the use of MMSE as its Bayesian analog and a simplistic choice. It also addresses the major drawbacks of conventional logCPM. I have added clarifications in Lns 57-68. The derivation details are included in Section 2.1 of Supplementary File 1.

7) The two summary statistics are actually correlated!! (L63 and see method) How would this affect the linear model fitting?

I would like to thank Reviewer #1 for noticing the technical detail. Their correlation is weak and well-expected. Because the linear model aims to estimate the coefficient alpha (Fig 1c) and test whether alpha=0, these aims are not affected by the correlations within covariates C themselves.

8) In L347 “For outlier removal, inverse of (estimated) cell variance was modeled with normal distribution” please use simulation and real data to demonstrate that the assumption is valid. See below.

9) In L 350-351, “Cells below the given Bonferroni adjusted P-value threshold were regarded as outliers in the next iteration”. The number of cells is usually large, Bonferroni correction is too conservative. You need to use appropriate way for multiple tests comparison.

I would like to thank Reviewer #1 for confirming the validity of assumption. Outlier removal is dataset-dependent and outcome-dependent. Here, outlier removal was only applied on the MARS-seq dataset, whose variance distributions before and after outlier removal were already demonstrated in Fig S13, which justified the whole outlier removal process and outcome.

10) What forward stepwise selection method is used? The threshold or parameter setting?

Nonlinear cellular summary covariates underwent restricted forward stepwise selection. The selection goal was best null co-expression P-value distribution (towards uniform distribution) instead of a regression problem. One nonlinear covariate was added at a time to the set of nonlinear covariates till the selection goal did not improve. It did not contain any threshold or any parameter other than the set nonlinear covariates.

11) L359, “The most enriched GO term (by P-value) was manually examined for contextual relevance and house-keeping role”. Please provide more objective steps rather than manual examination for readers.

I would like to thank Reviewer #1 for requesting extra details. The level of GO pathway removal depends on the biological question and context and can only be determined by the user. Imagine the co-expression network of dysfunctional T cells on the same dataset is studied for i) its selective (dis)advantage due to housekeeping activities and homeostasis, and ii) the relation between its dysfunction and co-receptor genes. i) would need minimal GO pathway removal but ii) would prefer the same level as in this manuscript. The computer program cannot judge user intention or distinguish these two scenarios. Therefore, a fully objective decision making is not possible.

I have added a sentence to clarify the criteria in this study (Lns 425-426):

Here we stop the iteration when the GO term is specific to the cell type (dys)function (immune function for dysfunctional T cells).

12) Sometime the authors used FDR, Q-value, or Bonferroni correction. Please explain clearly and be consistent.

I appreciate Reviewer #1’s comment on consistency. This study consistently used Q-values from BH procedure for FDR control from start to end. Other measures were also used when the goal was not controlling the FDR of a single set of exact hypothesis tests, such as with Bonferroni adjusted P-values below:

1. To compare positive hits within CRISPRi experiments or between CRISPRi experiments and against bulk experiments, Bonferroni adjusted P-values are more conservative to reduce false positives here and the downstream comparison error.
2. To aggregate multiple hypothesis tests from different gRNAs targeting the same gene when searching for its regulatory effects on other genes, it is unclear how the BH procedure or Q-value can achieve that, especially that each gRNA can have different off-target effects.
3. To control the FDR with permutation-based P-values of Gene Ontology enrichment on co-expression network, imprecision and ties in permutation-based P-values may affect the accuracy of BH procedure.

Additional clarifications are added in Lns 520-525.

13) Can the DE analysis be extended to a more general case, i.e., not only case-control?

I would like to thank Reviewer #1 for suggestions for potential extension. Indeed, I expect it to be extensible to DE against continuous variables or with donor kinships, as already mentioned in Discussion. These are promising future directions to explore but beyond the scope of this manuscript.

14) It would be helpful for the readers if the authors can provide a (cluster) visualization on the normalized expression as an intermediate output from Normalizr, since the possible confounders/covariates are already taken into account.

I would like to thank Reviewer #1 for suggesting this intuitive visualization. I have added Fig S9 and Lns 170-172, 506-509 for this analysis. Normalizr indeed removed library size confounding in UMAP embedding. Only Normalizr could recover the absence of cell population structure in the synthetic dataset of homogeneous cells.

Minor concerns:

In Figure 1 b) the color of read counts should be grey, same as the one in a).

In Figure 1 b) the color of normalized expression should be same as the one (normalized expression) in a)

I would like to thank Reviewer #1 for pointing out the mismatch. I have corrected their colors in Fig 1b.

L 477, “($Q \leq 0.05$ or 10^{-5})” why?

I appreciate Reviewer #2 for suggesting the ambiguity. These correspond to different cutoffs used in Fig 3h and Fig S12. I have clarified it in Lns 566-567 as below.

$Q \leq 0.05$ or 10^{-5} as specified in figure legend

L487, why did you use $P \leq 0.001$ but used Q-value for L484 and L485?

See below.

L488, the absolute value of logFC should be used.

The P-value cutoff is not for controlling the false discovery but instead for rejecting the alternative hypothesis. The aim of these conditions is to avoid identifying the scenario: intended target |--- gRNA ---| another cis-gene - another gene, which would otherwise become a false positive of gene regulation: gRNA ---| intended target - another gene. This motivation is now mentioned in Lines 581-583. The absolute value is not used because gRNA directly inhibits gene expression. Upregulation is unlikely to be a direct effect of gRNA. Also, following suggestions from Reviewer #2, I have expanded the relevant results section (Lns 230-241) and added references (Ref [41,21,22]) for causal inference theories behind this analysis.

Why is binomial approximate of multinomial? In line 57

Multinomial distribution can be approximated by a set of independent binomial distributions with the same probabilities p . This is an approximation because the sum of positive trials across all binomial distributions may differ from the total trial count in multinomial distribution.

Reviewer #2 (Expertise: CRISPR screening, statistical methods):

The author recognizes that the cell-to-cell technical variability and low read counts are two main challenges in analyzing single-cell RNA-seq data. To overcome this, they anticipate a potential unified two-steps that first normalizes data to rule out nonlinear confounding and then infers association between gene expressions at a single cell resolution using linear models. The author suggests that their method outperforms other existing methods of single-cell RNA-seq data analysis by accessing direct evaluation of P-values (from conditional Pearson correlation tests) and demonstrating extremely short computation time compared to alternative approaches. The author applies this method to identify differentially expressed genes and gene co-expression both from standard scRNA-seq datasets and from single-cell CRISPR experiments. This is an inferential method that takes a novel approach to identifying and correcting known and suspected confounder covariates prior to assessing statistical associations.

Overall, I find the method sound from a statistical perspective. However, the manuscript would benefit from simpler language and more layman explanations on some technical details and on the reasoning behind experiments shown. As is, I worry the manuscript is sub-setting itself to a more specialized audience. In addition, the manuscript would benefit from additional information regarding the details of the datasets used and reasoning behind choices made during the method development. Finally, I am not fully convinced this method outperforms existing methods and suggest a more thorough evaluation using standardized datasets and evaluation frameworks described in reference [8].

I would like to thank Reviewer #2 for their overall positive remarks and constructive feedbacks. To address the shortcomings and missing details suggested by Reviewer #2, I have extended the description of and the reasoning behind the choices of specific datasets, methods, and analyses. Below, I have also described the initial unsuccessful attempt in using the evaluation framework suggested by Reviewer #2, that ultimately led to the development of independent evaluations in this manuscript, which has been expanded following Reviewer #2's request.

Major points:

1. I have the following questions about the approach, which would benefit from more explanation in the text:
 - a. Minimum mean squared error estimator is a Bayesian method that requires information about the prior to estimate the posterior distribution. Could the authors be explicit on the specific parameters tested/assumed? It would be interesting to know how well the Bayesian approach compares to deterministic methods such as uniformly minimum variance unbiased estimator or a minimum variance unbiased estimator directly?

I would like to thank Reviewer #2 for the inspiring discussion. Here a non-informative (uniform) prior is applied so the posterior distribution only depends on the read counts of each gene in each cell, and the total read counts for each cell. This was documented in the Methods section and is now mentioned in the main text (Lns 57-68) for clarity and with additional justification in estimator choice.

Regarding the uniformly minimum variance unbiased estimator (UMVUE), in fact an unbiased estimator does not exist for $\log p$ in $\text{Binom}(n,p)$ although the UMVUE for p is trivial. The proof is included in Section 3.1 of Supplementary File 1. Note that using \log UMVUE of p is equivalent with $\log\text{CPM}$ with constant=0 and diverges for zero-read genes.

b. Modeling nonlinear effects with a Taylor Expansion is appropriate, but it would be informative to know what other non-linear methods were tested in modeling? Maybe there could be a quantitative comparison of these methods?

I would like to thank Reviewer #2 for suggesting the comprehensiveness. In fact, Taylor expansion appeared as the natural choice and therefore was the first and only approach attempted in the study. Given its already advantageous and satisfying performance, there was no motivation in trying any other for method development.

c. 'In contrast, edgeR and MAST, the best single-cell DE methods benchmarked in [8], had expression-dependent, 0- or 1-biased null P-values, which would lead to high false positive and false negative rates in application'. While the statement of 0 or 1 biased null p-values leading to higher FPR and FNR is seemingly self-evident, the authors should show data to demonstrate this explicitly if they wish to make this statement.

I would like to thank Reviewer #2 for the rigorousness. I have removed the claim on higher FPR and FNR (Ln 123).

2. The MARS-seq dataset is used to show generalizability of the method, and the author concludes from this analysis that the results are consistent across multiple scRNA-seq protocols. However, there is no description on how this protocol differs from the Perturb-seq protocol or how the data was processed to make it consistent with that from the Perturb-seq experiments.

I would like to thank Reviewer #2 for suggesting a self-contained manuscript. Details for each dataset are added in Lns 166-170 following the Minor point for Line 81. Description of data processing for quality control was already characterized in Methods (Lns 585-592). No other processing was needed to make it consistent with the Perturb-seq experiment.

In general, I find the evaluations of the ability of different methods to call differentially expressed genes somewhat limited. The Conquer dataset described in reference [8] has been specifically established to compare scRNA-seq pipelines by providing a repository of consistently processed, analysis-ready scRNA-seq datasets. That work also provides a clear framework that can be used to compare the performance of scRNA-seq pipelines. The author should take advantage of both to benchmark Normaliser against other tools in terms of: the number of genes that are found to be differentially expressed, the FPR, TPR, and AUROC. The author should also evaluate the robustness of their method to sample size and to UMI versus full-length datasets.

I would like to thank Reviewer #2 for this suggestion. The Conquer pipeline was indeed my initial but unreported benchmarking attempt when the study was initiated. Several benchmarks and softwares benchmarked in this manuscript were motivated by Ref [8]. However, I experienced several major limitations of Conquer in both the pipeline and the dataset, and eventually reasoned that a separate benchmarking would be more realistic, relevant, and informative:

1. Its pipeline is not actively maintained and could not be run. The software “powsim” for simulating benchmark datasets was incompatible with newer Bioconductor versions. The updated software “powsimR” was then incompatible with the Conquer pipeline. The author stated the repository “is not intended to be a software package or a general pipeline for differential expression analysis of single-cell data”. See the discussion between the author and another researcher with similar experiences: https://github.com/csoneson/conquer_comparison/issues/5 .

2. The datasets contain much fewer cells than a modern UMI-based experiment. The majority of differential expression benchmarks were performed with around or less than 100 cells. Even the largest cell count for differential expression benchmarking was much smaller than today’s typical UMI-based experiments. These benchmarks are unlikely to reflect real performance with modern UMI-based datasets.

3. Its benchmark statistics are in fact limited and could not reveal sufficient details to inform normalization method development and choice. For example, the null P-value bias by edgeR is highly dependent on expression level (Fig S4), which at the global level persists but becomes hidden (Ref [8]). FPR/TPR values at individual P-value cutoffs or nominal FDRs cannot capture their curves at varying cutoffs. Ref [8] also lacks many benchmarks such as co-expression on synthetic null datasets (Fig 2de) or differential expression on simulated or real positive controls (Fig 2fgh).

In total, I believe the separate benchmarking in this manuscript 1) uses modern and realistic simulations (including Splatter and SymSim suggested by Reviewer #1), 2) is more relevant with modern data sizes, and in fact 3) provides more comprehensive benchmark statistics than Conquer.

I would like to thank Reviewer #2 for proposing full-length datasets. UMI-based platforms have mature commercial solutions that can scale to many (10K or more) cells typically more easily than full-length platforms. Most analyses in this study, such as high-MOI CRISPR scRNA-seq and cell-type-specific co-expression network, need this many cells in total, more than a typical experiment with popular full-length protocols. Full-length measurements also follow different read count distributions from UMI based measurements (e.g. with zero inflation, Ref [57]) which may need additional validation or normalization steps. Therefore, this study focuses on UMI-based datasets, with clarification added in Lns 324-328.

Variations in sample size indeed can improve evaluation robustness. Actually, most evaluations already covered a wide range of effective sample sizes, which were determined by the random grouping or gRNA assignment. This reflected real-world scenarios such as marker gene discovery for abundant to rare cell types and single-cell CRISPRi screen. In simulations with SymSim and Splatter (suggested by Reviewer #1), cell counts from the minor group of DE also varied from 519 to 3,927. I have added a new paragraph to clarify that in Lns 159-165.

One of the motivations to develop this novel method is that “A comprehensive benchmarking found single-cell-specific attempts in DE could not outperform existing bulk methods such as edgeR [8]”. However, it is not clear to me that Normalizr achieves the goal of outperforming edgeR, particularly as edgeR is used as the “ground-truth” in many of the analysis performed (see below).

I would like to thank Reviewer #2 for pointing out the ambiguity. The only analysis that used edgeR as a proxy for ground-truth was the logFC estimation in positive controls of Perturb-seq, because the true logFC was not known. To avoid the ambiguity and to highlight objective comparisons, I have 1) replaced positive control P-value comparisons against edgeR (original Fig 2h, S6a) with comparisons among all methods (current Fig 2h), and 2) removed positive control logFC comparisons against edgeR (original Fig 2h, S6b) because the true logFCs are unknown. LogFC estimation evaluation is currently supplemented with synthetic datasets of null co-expression and from SymSim and Splatter (as suggested by Reviewer #1) in Fig 2fg, S5, S8, and Lns 133-141. With the current extended evaluations suggested by Reviewers #1 and #2, Normalizr demonstrates superior performance than edgeR in all the evaluations except being comparable with edgeR in logFC estimation (Fig 2, S2, S4-S9).

Minor points:

Specific points about the datasets used/analysis performed that would benefit from clarification:

Line 81: “We downloaded a high multiplicity of infection (high-MOI) CRISPRi Perturb-seq dataset of K562 cells [6]” Which dataset is this? From the materials and methods of the paper you reference it seems all Perturb-seq experiments are performed with a low MOI. Specifically, authors from that paper list only 3 single-cell CRISPR experiments (and corresponding infection rates):

- “pilot experiment” - transduction efficiency of 20%–30%
- “UPR epistasis experiment” - transduction efficiencies of 5%–10%
- “UPR Perturb-seq experiment” - transduction efficiency of 15%

All of these correspond to low, not high MOIs. Additionally, can you provide in the materials and methods the links from which the datasets were downloaded? Are you starting your analysis from raw counts? Unmapped reads? In addition, writing a summary on how these datasets were generated and why you selected them to benchmark your method would help to give context to the analysis you are presenting, and not force readers to go to the original publication

I very much appreciate Reviewer #2 for catching this mistake and providing these constructive feedbacks. This is the UPR Perturb-seq experiment which is now corrected in Ln 90. Data accession numbers were already included in Data availability and Code availability sections (Lns 611-617). This study always uses published UMI read count and gRNA assignment matrices which is clarified in Lns 90-91, 533-535, 585-592.

Line 95: how many cells have no detectable levels of gRNA expression? What is the size of your random groups? In the same line: what is the size of your “10 equally sized” gene subsets? How do you define the expression level bins?

I appreciate Reviewer #2’s attention to details. 4,622 cells did not contain detectable gRNA, as previously included in Tab S1 and now added in Ln 114. Each random partition had a different partition rate, sampled uniformly between 5% and 50%, as included in Methods (Lns 431-432). The size of each of the 10 equally size gene subsets is 976 or 977, computed from Tab S1 and indicated in the legend of Fig 2a. The expression level bins are based on the proportion of expressed cells, as detailed in the legend of Fig 2a and now also in Lns 118-119.

Line 115: how is the “ground-truth logFC” defined?

I would like to thank Reviewer #2 for pointing out the missing detail. This is defined as the difference in mean log expression proportion, as clarified in Lns 482-484. Ground-truth logFC for SymSim and Splatter is defined in Lns 489-490.

Line 127-129: how are you measuring sensitivity here? Are you taking single cells expressing each of the gRNAs and checking if the targeted gene is significantly downregulated? How does this look at the level of individual cells? Specifically, how often for each method do you find that the targeted gene is significantly downregulated? In what fraction of cells does each method ‘fail’ to identify your true-positive gene? In supplementary Figure 6 you compare edgeR with several other methods. It would be more important for the purpose of validating your method to compare Normaliser with all other methods.

I appreciate Reviewer #2 for noting the lack of details. This comparison on sensitivity or statistical power is evaluated on the distribution of P-values from hypothesis testing of positive controls in Fig 2h. This is fairer than the evaluation Reviewer #2 suggested that accept per-cell expression levels as input before hypothesis testing because 1) the latter need to define a test statistic (e.g. Wilcoxon rank-sum test), maybe implicitly, which themselves can be underpowered or biased toward one method over another, and 2) the latter cannot include DE methods such as edgeR, MAST, and Seurat. The P-value distribution also shows the full picture without depending on the choice of significance threshold. We also included a single number to characterize overall method sensitivity as the average log P-value of positive controls (Fig 2h). Both the P-value distribution and the average log P-value showed a clear sensitivity advantage from Normaliser compared to other methods with unbiased P-values.

I have rephrased the paragraph in Lns 149-156 to clarify the analysis. Also following the final Major point, I have removed direct comparisons against edgeR to avoid confusion. Fig 2h now objectively

groups and compares methods that have unbiased P-values (Fig 2b), because biased P-values could not reflect statistical power.

Line 132-134: It is rather strange to me that edgeR logFCs are being used as the ground-truth for your analysis, particularly since in Figure 2b you showed that in the null synthetic dataset it did not return a uniform null distribution of pValues and suggested that this could contribute to high false positive and false negative rates. Can you please explain your reasoning further?

Addressed in the last Major point.

In addition, you write: “Normaliser also estimated logFCs more accurately than other normalization or imputation methods when using edgeR as a proxy for ground-truth, which performed best in the synthetic dataset” Where is this statement quantified?

This was quantified in Fig 2h (previous version) but is now removed along with this statement to address the ambiguity.

Line 144-146: Your false positive analysis is based on the hypothesis that cells may receive more than one gRNA leading to “competition between gRNAs for sCas9 or limited read counts”, what is the evidence that this is actually the case? In addition, how many cells do you find in this dataset where with more than 1 assigned gRNA? What is the average gRNA number per cell? How many of those have NTC gRNAs?

I wish to thank Reviewer #2 for pointing out the potential ambiguity. To clarify, it studies the consequence of gRNA cross-association, instead of gRNA competition. There are many potential sources of gRNA cross-association but gRNA competition is only mentioned as one possible source. This study does not determine its origin but demonstrates its existence and consequence which can be accounted for by Normaliser. I have rephrased Lns 178-181 to address the confusion. The evidence for gRNA cross-association is in Fig 3b and the evidence for elevated FPR is in Fig 3cd, S10.

Over 98.7% of cells contained more than one gRNA, with 30 gRNAs per cell on average. Over 20% cells contained at least one of 101 NTC gRNAs. These are now detailed in Lns 185-186.

Line 156-157: “Notably, the FPR was significantly higher among genes whose transcription start sites (TSS) were targeted by another gRNA” Where is the data supporting this statement?

This is supported by different FPRs in Fig 3c and its legend. I have added the FPRs in the main text to indicate the location (Lns 193-194).

Line 173-174: please provide details about what the “small screen”. How many gRNAs? How many targeted genes? Are gRNAs designed against the TSS? What is the difference between this screen and the previous one?

I appreciate Reviewer #2’s suggestion on manuscript improvement. I have updated the description in Lns 187-189 as quoted below:

This was reproducible in a small-scale pilot screen of the same study with 3,117 gRNAs and 47,650 cells averaging at 18 gRNAs per cell, including the same NTCs and TSS-targeting gRNAs but fewer enhancer-candidate-targeting gRNAs.

Line 186-187: can you expand further on the analysis you are doing here and why? Are you looking for downstream effects of knocking down a gene to try to understand some of its functions?

I would like to thank Reviewer #2’s helpful request for motivation. This is an analysis to screen for gene regulations and reconstruct gene regulatory networks, which is a question on its own in systems biology. I have expanded the paragraph to describe the question and theory in Lns 230-241.

Line 273: “However, Normalisr is not designed for nonlinear associations”. I understand the author's sentiment, but this is confusing wording as your method does deal with nonlinearities. I appreciate Reviewer #2's suggestion of the confusion. I have rephrased it to “Normalisr is not designed for hypothesis testing of arbitrary nonlinear associations” in Ln 321.

Reviewer #3 (Expertise: gene co-expression networks (using scRNASeq data))

The manuscript proposes a method named Normalizr to detect DE or co-expressed genes from single-cell count data. A major advantage of Normalizr is its computational efficiency compared with existing tools. Normalization is an important procedure in single-cell analysis, and if properly used, can help increase the accuracy of downstream statistical analysis. However, I have several major concerns about the statistical validity of the proposed procedures in Normalizr. The statistical tests for differential expression and co-expression remain questionable before these issues can be resolved.

I very appreciate Reviewer #3's acknowledgement on the importance of this direction and their effort in going through the technical details and pointing out the missing links. Following Reviewer #3's feedbacks, I have revised the manuscript to include these details and responded to Reviewer #3's concerns with extra clarifications below.

The title highlights the application of Normalizr on single-cell CRISPR data but the method itself seems to be designed for single-cell RNA-seq count data in general. Are there any specialized considerations in Normalizr for better performance with CRISPR data?

I wish to thank Reviewer #3 for acknowledging the general purpose of Normalizr. I do expect it to have wider applications but only to be demonstrated in future studies, such as 'soft' groupings for DE and kinship-aware population studies mentioned in the conclusion. This manuscript is focused primarily on single-cell CRISPR screen analysis and secondarily on co-expression because:

1. CRISPR datasets are more direct evidences of causal effects in gene regulation than co-expression and therefore may carry stronger mechanistic insights in biology.
2. Single-cell CRISPR screen is an emerging and powerful experimental technology which however lacks a suitable computational/statistical method. No existing method can estimate effect size and perform hypothesis testing efficiently and statistically soundly. In contrary, there have been multiple studies of single-cell co-expression networks with bulk or single-cell methods where hypothesis testing is less emphasized.
3. The single-cell CRISPR dataset contains positive and negative controls that can directly evaluate inference results.

A major question is about the concept of logCPM. It refers to log-transformed CPM that can be directly calculated using the count matrix. Why does logCPM need an estimator? If logCPM refers to a random variable instead of observed statistics in this manuscript, it's better to replace it with a new term.

I would like to thank Reviewer #3 for pointing out the ambiguity. Bayesian logCPM is now renamed as Bayesian log expression. Its mathematical notation $lcpm$ is also replaced with y or Y .

Also, how is the MMSE estimator obtained? This is an important step but not mentioned in the manuscript. A follow-up question is why the Bayesian estimator instead of observed read counts is used to calculate confounding effect? As the Bayesian estimator only uses a non-informative prior, this estimation step does not seem to offer any advantage.

I wish to thank Reviewer #3 for this question. I have added a detailed derivation of the MMSE estimator in Section 2.1 of Supplementary File 1. The confounding effects (cellular summary covariates) are indeed computed from observed read counts rather than the Bayesian estimator, with clarifications in Lns 73-74 and Section 1.1 of Supplementary File 1.

One theoretical motivation for MMSE estimator for log expression is that the uniformly minimum variance unbiased estimator (UMVUE) does not exist, as discussed in the reply to Reviewer #2. Note that $\log(\text{CPM} + \text{constant})$ relies on the UMVUE of expression instead of log expression, so the downstream analyses are biased. The motivation of using MMSE estimator and not using an informative prior is now detailed in Lns 57-70. The MMSE estimator provided obvious advantages over $\log(\text{CPM} + \text{constant})$ which relies on the UMVUE of expression, as evidenced in the improved removal of technical confounding in Fig S2, S7.

For the estimation of confounding effects, the notations involve the letter c in uppercase, lowercase, and mathematical font, and are very confusing. It should be helpful to label corresponding notations in Fig 1b.

(1) What are the definitions of functions $C(c)$?

(2) It seems that Taylor expansion step should be called a polynomial regression instead. Please specify which terms are observed and which are parameters in need of estimation in a clearer manner. Please also explain how parameter are estimated in this step.

(3) Is the optimal candidate covariate in line 312 simply selected between log total read count and number of 0-read genes?

I appreciate Reviewer #3's effort in navigating through the mathematical details. I have rephrased this section with updated annotations, and included the notations in Fig 1b to address the confusion.

(1) The functions $C(c)$ are generic unknown functions. After showing that we can use their Taylor expansion terms instead, we restricted $C(c)$ to individual polynomial terms. The restricted forward stepwise selection found three important $C(c)$ functions: log total read count, its square, and the number of 0-read genes. This is now detailed in the section (Lns 350-378).

(2) Polynomial regression is indeed similar but typically refers to estimating/testing the nonlinear effect of the variable instead of removing them as covariates. This is a feature selection question that does not fall into the traditional class of statistical questions such as point estimation. The terms observed are described in Lns 357-359. The features to select are described in Lns 366-368.

(3) In each iteration, the optimal candidate covariate is selected from the set in Eq 2 (Lns 370-371).

This contains only log total read count and number of 0-read genes in the first iteration, but also grows to include some of their polynomial combinations in the subsequent iterations.

Line 332: Why the gene expression is transformed as described in line 333 instead of directly using a Z score format statistic? What's the rationale of using scaling factor γ_i ? This step seems quite arbitrary. As number of zero counts are already accounted for when estimating confounding effect, what's the purpose of including it again in variance normalization?

A related question is, assuming confounding effects can be identified and removed after regression, why these effects are still included in the linear models (lines 340 and 343) to test for differential and co-expression?

I appreciate Reviewer #3's effort in navigating through the normalization steps. It is unclear what "a Z score format statistic" means here. Z score loses true variance information and cannot estimate effect sizes such as logFC. Here we only normalized the variance while maintaining the mean, which matches with the definition of this step as variance normalization. Normalizing the variance while removing the mean would work identically because the mean is also removed in the linear association testing step. However in either case the covariates are still needed in the linear association step because neither the binary grouping in DE nor other covariates such as untested gRNAs are orthogonal to existing covariates.

The motivation of scaling factor γ_i is already described in this section (Lns 394-396).

Specifically, technical variations in scRNA-seq form a major proportion of expression variance, especially in lowly expressed genes which need variance normalization. Conversely, the variance of

highly expressed genes are more accurately measured like in bulk RNA-seq, and should not be normalized. The use of γ_i can account for the difference in variance composition and demonstrate advantage in logFC estimation (Fig 2fg, S5, S8).

In the simulation of null dataset, a key issue is whether the synthetic data captures the confounding effect between read counts and covariates such as log total read count or 0-read gene count. If the synthetic data fails to capture such relationship, the p values from these data would not be very useful. I would like to thank Reviewer #3 for suggesting the sanity check. The existence of confounding is demonstrated on conventional logCPM or existing methods in Fig S2, S7. I have added texts in Lns 99-102 to confirm its confounding effect.

Line 64: This sentence is confusing. Please provide more details about what the L0 and L1 norms refer to in this context.

I would like to thank Reviewer #3 for requesting more details. The L0 and L1 norms are on read count vector (Lns 73-74). I have added Supplementary File 1 to clarify that (in Section 1.1).

Line 293: Please provide a formula for the non-informative prior.

I wish to thank Reviewer #3 for pointing out the missing details. I have clarified the non-informative prior to be a standard uniform distribution in Ln 346, and detailed the derivations in Section 2.1 of Supplementary File 1.

Results:

Fig 2a. In addition to the p value densities, can the author directly show the false positive rate of Normalizr in the negative control study, compared with other methods?

I appreciate Reviewer #3's suggestion on the visualization. I have added extra panels for quantile-quantile plot (Q-Q plot) of null P-values in Fig 2a, S4, S6. Q-Q plot shows the false positive rate (Y) at different cut-offs (X). The conclusions did not change with this new analysis.

Fig 2e. Most methods listed here were not designed for studying co-expression. How were these methods used to obtain co-expression p values? It is important to provide more details for a fair comparison.

I would like to thank Reviewer #3 for the clarification request. The same hypothesis test (Fig 1c) was performed on the values after any normalization or imputation to ensure a fair comparison. Extra clarification is added in Lns 126-127.

The negative control study with the Perturb-seq data is helpful, but it would be more informative to add a negative control based on single-cell data from at least one another platform, to help exclude platform-specific factors.

I would like to thank Reviewer #3 for suggesting comprehensiveness. All possible evaluations were already replicated for the MARS-seq dataset in Fig S1b, S3b, S6-9. Compared to the Perturb-seq dataset from an *in vitro* 10X cell-line study, the MARS-seq experiment was performed on frozen cancer tissues, with prior antibody-based fluorescence-activated cell sorting (FACS), at a much lower library size, and on a well-based platform. Also following suggestions from Reviewer #2, I have added more details to emphasize their platform differences in Lns 168-170.

Fig 3b. What's the difference between the full-scale and small-scale data?

I appreciate the request for details from Reviewers #2 and #3 on the datasets. The small-scale experiment was a pilot experiment of the same study with fewer cells and gRNAs per cell, as now described in Lns 187-189. The NTCs and TSS-targeting gRNAs are identical.

Fig 3h. How was the off-target rate obtained?

I would like to thank Reviewer #3 for raising this uncertainty. I have updated the brief description for off-target rate in Lns 223-226 (quoted below) and the technical details in Lns 565-574.

With two gRNAs targeting the same TSS, we quantified the proportion of off-target effects as the proportion of significant associations between the weaker gRNA (in association with the targeted gene) and its trans-genes (over 1Mbp away from the targeted site or on different chromosomes) that were highly insignificant for the stronger gRNA.

Reviewer #4 (Expertise: scRNASeq, Crispr screening):

The author presents a unified framework for normalisation and statistical association testing for single-cell CRISPR screens. Compared to the existing literature, this method (Normalisr) contributes with the following features:

- 1) a normalization step that models nonlinear confounding effects on gene expression by iteratively introducing higher order covariates;
- 2) improved sensitivity, specificity, and efficiency in the normalisation framework;
- 3) an association step that utilizes linear models to unify case-control DE, single-cell CRISPR screen analysis as multivariate DE, and gene co-expression network inference;
- 4) quantification of off-target effects;
- 5) higher computational efficiency.

The author demonstrates applications of this method in two scenarios: gene regulation screening from pooled CRISPRi CROP-seq screens and the reconstruction of transcriptome-wide co-expression networks from conventional scRNA-seq.

The method was tested by performing a wide range of evaluations and compared to other existing, relevant methods.

I believe that the contributions mentioned above are a valuable addition to the existing literature.

The method is well documented on GitHub; I could easily install the package and run the two examples:

```
normalisr/examples/GSE120861  
normalisr/examples/GSE123139
```

I report few comments / questions below.

I wish to thank Reviewer #4 for the positive assessment of the manuscript and particularly the software. I have improved the manuscript with additional clarifications to address each of Reviewer #4's comments, especially the new schematic Fig 5 to highlight Normalisr's features and functionalities in comparison with other methods.

Line 29: "A comprehensive benchmarking found single-cell-specific attempts in DE could not outperform existing bulk methods such as edgeR".

Line 132: "Normalisr also estimated logFCs more accurately than other normalization or imputation methods when using edgeR as a proxy for ground-truth, which performed best in the synthetic dataset".

Comment 1) The reader might wonder what is the point of developing a single-cell-specific method in DE if this person can just use a method for bulk. I believe that this issue should be discussed more clearly in the text.

I appreciate Reviewer #4 for suggesting the ambiguity. EdgeR only had a minor advantage in logFC scale estimation while Normaliser outperformed edgeR in all other metrics including FDR control, speed, and logFC variance (Fig 2). Also following the feedbacks from Reviewer #2, I have removed the analysis that used edgeR as a proxy for ground-truth (old Ln 132) which may falsely imply optimal performance from edgeR. I also added extra clarifications for the limitations of existing methods (Lns 34-35) and for Normaliser's superior performance (Lns 172-175).

Line 95: "We partitioned cells that did not detect any gRNA into two random groups 100 times to evaluate the null P-value distribution of single-cell DE".

Comment 2) Why didn't the author use cells assigned to non-targeting control guides rather than partitioning cells based on lack of detection?

I have added a sentence to explain the choice (Lns 116-117):

This provides better homogeneity than using cells with non-targeting control gRNAs (NTCs) because they may have unknown and unintended targets (see below).

Line 485: "Significant association with trans-gene: $Q \leq 0.2$, $|\logFC| \geq 0.05$ "

Comment 3) The chosen threshold is barely significant.

I appreciate Reviewer #4's attention to details. First, $Q \leq 0.2$ in a single test means that 80% of the predictions are expected true positives. Moreover, each regulation needed to satisfy these criteria four times, for both gRNAs in both datasets (Ln 576). Together this is a strong threshold. This is also clarified in Ln 242. LogFC was thresholded at a low number to account for gRNAs with low efficiency.

Comment 4) Supplementary Figure 7 is reported in 4 pages, Supplementary Figure 12 in 3 pages. I would split figures and legends into, possibly, a single page dedicated to each figure.

I would like to thank Reviewer #4 for the visual improvement. I have split Fig S7 into Fig S1b, S3b, S6, S7, and S8, and Fig S12 into Fig S14 and S15.

Comment 5) Maybe the author could summarise and highlight in a table / schematic the features that Normaliser has and that the other methods don't have.

I am grateful for Reviewer #4's suggestion, which has resulted in Fig 5.

Reviewers' Comments:

Reviewer #1:

Remarks to the Author:

Thank the authors' efforts to address the comments well. Only some minor concerns:

1. The new method is missed in a) of Figure S5.
2. Add the explanation in the main text -
"The two summary statistics could be correlated. However, these correlation do not affect ... because ..."
3. In Figure S13, the scales of y-axis need to be same for a good visual comparison between the two distribution patterns.
4. Regarding to the forward stepwise selection, you mentioned that "One nonlinear covariate was added at a time to the set of nonlinear covariates till the selection goal did not improve." What is the measure of the improvement?
5. Suggestion: move Figure S9 to the main text as it can give readers a direct impression of the comparisons.

Reviewer #2:

Remarks to the Author:

I congratulate the author for a much improved manuscript. The reasoning behind the selection of specific datasets for the benchmarking of the Normalizr software and the specific analysis performed are now much clearer making this work more likely to resonate with a wider audience.

In addition, the author has appropriately addressed all my comments and concerns, and I believe the work is suitable for publication.

One minor suggestion I would add (which I decided not to raise in my first review): I agree with reviewer #3 that the title could be modified to better fit the actual method described. I understand the desire to include "CRISPR" in the title as this is a very active area of research. However, to highlight that this is a general method for normalization of single-cell data that can be applied to CRISPR screens I suggest changing it to something along the lines of:

"Normalizr: normalization and association testing for single-cell data including CRISPR screens and gene co-expression analysis"

Reviewer #3:

Remarks to the Author:

I would like to thank the author's efforts in responding to my comments. The current manuscript is clearer and has provided additional evidence, but some of my previous questions about the method remain unresolved:

It is still not clear what variables are in $C(\text{orig})$ in addition to log total read count and the number of 0-read genes.

A more intuitive explanation of formulas (1) and (2) is needed so that the methodology can be understood by more general audience.

What does the "cell error variance" refer to? Why gene expressions are normalized using a cell-wise variance factor instead of a gene-wise factor?

Reviewer #4:

Remarks to the Author:

The author addressed all of my points and I do not have any further comments to make.

I would consider the manuscript suitable for publication, provided that a thorough response to the remarks raised by the other reviewers is given.

I very appreciate all Reviewers' valuable time and contribution in improving this manuscript through their prompt, positive, and constructive evaluations of the last revision. Your concerns continue to point out the directions this manuscript can further improve along. Following Reviewers' feedbacks, I have updated the manuscript title, upgraded one figure to the main text, improved the text, and attempted new visualization styles as reported in blue below. The manuscript updates are also highlighted in blue.

REVIEWER COMMENTS

Reviewer #1 (Remarks to the Author):

Thank the authors' efforts to address the comments well. Only some minor concerns:

1. The new method is missed in a) of Figure S5.

I appreciate Reviewer #1's attention to details towards a better manuscript. The seemingly missing panels for Normalizr in Fig S2, S4, S5 are already included in Fig 2 and not included here to avoid duplication. I have updated their captions for clarification.

2. Add the explanation in the main text -

“The two summary statistics could be correlated. However, these correlation do not affect because “

I would like to thank Reviewer #1 for suggesting the specific text edit. I have added the explanation after rephrasing to fit it into context and writing style in Lns 87-88.

3. In Figure S13, the scales of y-axis need to be same for a good visual comparison between the two distribution patterns.

I appreciate Reviewer #1's suggestion on improving visual comparison. Using the same scale for y-axis was indeed the straightforward approach and also the initial attempt before the current version. However, linear Y cannot show outlier cells before outlier removal due to their rarity and log Y hides the near-normal distribution in bell shape after outlier removal (right figure here).

Note that the ultimate aim of Fig S12 (previous Fig S13) was to demonstrate successful cell outlier removal, while direct visual comparison is only one potential approach to achieve that. The current version already clearly demonstrates several outlier cells deviating from the major population (Fig S12 top), and their successful removal with a near-normal distribution (Fig S12 bottom). Therefore, a good visual comparison is not necessary.

To address Reviewer #1's concern, I have revised the caption of Fig S12 to emphasize the conclusions and to avoid suggesting direct comparison.

4. Regarding to the forward stepwise selection, you mentioned that “One nonlinear covariate was added at a time to the set of nonlinear covariates till the selection goal did not improve.” What is the measure of the improvement?

I am grateful for Reviewer #1’s pointing out the missing detail in Methods. The selection goal here is the uniform distribution of null co-expression P-values. Measurement of improvement is less deviation from the uniform distribution (Fig S3). This has been mentioned in the main text in Lns 104-105, and is now also clarified in Lns 386-387.

5. Suggestion: move Figure S9 to the main text as it can give readers a direct impression of the comparisons.

I wish to acknowledge Reviewer #1 for sharing the optimism in the performance and potential utility of Normalisr. I have moved Fig S9 to Fig 3 with improved figure formatting. The related text has also been updated and expanded to match this main figure in Lns 171-182, 522-523.

Reviewer #2 (Remarks to the Author):

I congratulate the author for a much improved manuscript. The reasoning behind the selection of specific datasets for the benchmarking of the Normalizr software and the specific analysis performed are now much clearer making this work more likely to resonate with a wider audience.

In addition, the author has appropriately addressed all my comments and concerns, and I believe the work is suitable for publication.

One minor suggestion I would add (which I decided not to raise in my first review): I agree with reviewer #3 that the title could be modified to better fit the actual method described. I understand the desire to include "CRISPR" in the title as this is a very active area of research. However, to highlight that this is a general method for normalization of single-cell data that can be applied to CRISPR screens I suggest changing it to something along the lines of:

"Normalizr: normalization and association testing for single-cell data including CRISPR screens and gene co-expression analysis"

I wish to thank all four Reviewers for acknowledging Normalizr's generality/extensibility on different occasions throughout this reviewing process. Following your suggestions and encouragement, I have updated the title as:

Normalizr: single-cell data normalization and association testing unifying CRISPR screen and gene co-expression analyses

Reviewer #3 (Remarks to the Author):

I would like to thank the author's efforts in responding to my comments. The current manuscript is clearer and has provided additional evidence, but some of my previous questions about the method remain unresolved:

It is still not clear what variables are in $C(\text{orig})$ in addition to log total read count and the number of 0-read genes.

I wish to thank Reviewer #3 for exactly specify their doubt. $C(\text{orig})$ is indeed log total read count and the number of 0-read genes, as mentioned explicitly in Lns 359-360.

A more intuitive explanation of formulas (1) and (2) is needed so that the methodology can be understood by more general audience.

I appreciate Reviewer #3 for suggesting improvement for general readership. I have added their intuitive interpretations in Lns 370-371, 379-380.

What does the "cell error variance" refer to? Why gene expressions are normalized using a cell-wise variance factor instead of a gene-wise factor?

I would like to thank Reviewer #3 for suggesting the uncertainty. I assume Reviewer #3 meant "cell-specific error variance" in Cell variance estimation in Methods. This is standard variance normalization across samples (here cells) to remove confounding effects from known covariates. The complexity and potential confusion arose because variance normalization was not performed separately on each gene (i.e. variable), but instead on the total variance of all genes as a single variable in order to reduce overfitting.

It is unclear how a "gene-wise factor" according to Reviewer #3 is exactly defined, without which no concrete comparison is possible here. However, many of them (e.g. Z-score transformation) discard biological differences in variance levels, cannot estimate logFC, and most importantly are not designed to and cannot address variance confounding across samples.

For clarification of the exact approach and motivation, I have included additional details in Lns 395-404.

Reviewer #4 (Remarks to the Author):

The author addressed all of my points and I do not have any further comments to make.
I would consider the manuscript suitable for publication, provided that a thorough response to the remarks raised by the other reviewers is given.
I wish to thank Reviewer #4 for their evaluation.